# Lightweight and Interpretable Transformer via Mixed Graph Algorithm Unrolling for Traffic Forecast

Ji Qi [1]  Mingxiao Liu [1]  Tam Thuc Do [2]  Yuzhe Li [1]  Zhuoshi Pan [1]  Gene Cheung [2]  H. Vicky Zhao [1]

## Abstract

Unlike conventional "black-box" transformers with classical self-attention mechanisms, we build a lightweight and interpretable transformer-like neural network by unrolling a mixed-graph-based optimization algorithm to forecast traffic with spatial and temporal dimensions. We construct two graphs: an undirected graph $\mathcal{G}^u$ capturing spatial correlations across geography, and a directed graph $\mathcal{G}^d$ capturing sequential relationships over time. We predict future samples of signal $\mathbf{x}$, assuming it is "smooth" with respect to both $\mathcal{G}^u$ and $\mathcal{G}^d$, where we design new $\ell_2$- and $\ell_1$-norm variational terms to quantify and promote signal smoothness (low-frequency reconstruction) on a directed graph. We design an iterative algorithm based on alternating direction method of multipliers (ADMM), and unroll it into a feed-forward network for data-driven parameter learning. We periodically insert graph learning modules for $\mathcal{G}^u$ and $\mathcal{G}^d$ that play the role of self-attention. Experiments show that our unrolled networks achieve competitive traffic forecast performance as state-of-the-art prediction schemes, while reducing parameter counts drastically. Code: https://github.com/SingularityUndefined/Unrolling-GSP-STForecast.

## 1. Introduction

Transformer, based on the classical self-attention mechanism (Vaswani et al., 2017), is now a dominant deep learning (DL) architecture that has achieved state-of-the-art (SOTA) performance across multiple fields (Gillioz et al., 2020; Za-

mir et al., 2022). However, like other "off-the-shelf" DL models, transformer requires a massive number of parameters, and its internal structure is not easily interpretable. Model size reduction is a real industrial concern for practical computation- and/or memory-constrained environments[1].

*Algorithm unrolling* (Monga et al., 2021) offers an alternative hybrid model-based / data-driven paradigm; by "unrolling" iterations of a model-based algorithm minimizing an optimization objective into neural layers stacked together to form a feed-forward network for data-driven parameter learning, the resulting network can be *both* mathematically interpretable[2] *and* competitive in performance. Notably, Yu et al. (2023) designs an algorithm minimizing a sparse rate reduction (SRR) objective that unrolls into a transformer-like neural net for image classification. Thuc et al. (2024) designs graph-based algorithms minimizing graph smoothness priors that unroll into transformer-like neural nets for image interpolation while reducing parameters. However, only a positive *undirected* graph model was used in Thuc et al. (2024) to capture simple pairwise correlations between neighboring pixels in a static image.

*Spatial-temporal data* (*e.g.*, traffic and weather) contains more complex node-to-node relationships: geographically near stations exhibit *spatial correlations*, while past observations influences future data, resulting in *sequential relationships*. In this paper, we study the design of transformers via graph algorithm unrolling for spatial-temporal data, using traffic forecast as a concrete example application.

We roughly categorize traffic forecast methods into model-based and DL-based methods. Among model-based methods, one approach to traffic forecast is to filter the signal along the spatial and temporal dimensions independently (Ramakrishna et al., 2020)—*e.g.*, employ graph spectral filters in the *graph signal processing* (GSP) literature (Ortega et al., 2018; Cheung et al., 2018) along the irregular spatial dimension, and traditional Fourier filters along the

---

[1]Department of Automation, Tsinghua University, Beijing, China [2]Department of EECS, York University, Toronto, Canada. Correspondence to: H. Vicky Zhao <vzhao@tsinghua.edu.cn>, Gene Cheung <genec@yorku.ca>.

*Proceedings of the 43rd International Conference on Machine Learning*, Seoul, South Korea. PMLR 306, 2026. Copyright 2026 by the author(s).

---

[1]See a brief review of lightweight learning models for different applications in Appendix A.1.

[2]Common in algorithm unrolling (Monga et al., 2021), "interpretability" here means that each neural layer corresponds to an iteration of an optimization algorithm minimizing a mathematically-defined objective.

regular time dimension. Another approach is to model the spatial correlations across time as a sequence of slowly time-varying graphs, and then filter each spatial signal at time $t$ separately, where the changes in consecutive graph adjacency matrices in time are constrained by the Frobenius norm (Kalofolias et al., 2017), $\ell_1$-norm (Yamada et al., 2019), or low-rankness (Bagheri et al., 2024). However, neither approach exploits spatial and temporal relationships simultaneously for optimal performance.

Recent DL-based efforts address traffic forecast by building elaborate transformer architectures to capture spatial and temporal relations (Feng & Tassiulas, 2022; Gao et al., 2022; Jin et al., 2024)[3]. Like transformers used in other fields (Zamir et al., 2022), these are also black boxes with large parameter counts. *Graph attention networks* (GAT) (Veličković et al., 2018) and *graph transformers* (Dwivedi & Bresson, 2020) are adaptations of the self-attention mechanism in transformers for graph-structured data (*e.g.*, computing output embeddings using only input embeddings in local graph neighborhoods). However, they are still uninterpretable and maintain large parameter sizes.

In contrast, we extend Thuc et al. (2024) to build lightweight "white-box" transformers via *mixed* graph algorithm unrolling for traffic forecast. Our networks are interpretable in two ways: i) the unrolled neural layers correspond to algorithm iterations minimizing a graph-based objective, operating as low-pass graph filters; ii) our learned graphs reveal critical patterns that influence traffic dynamics.

Specifically, we learn *two* graphs from data: i) an *undirected* graph $\mathcal{G}^u$ to capture spatial correlations across geography, and ii) a *directed* graph $\mathcal{G}^d$ to capture sequential relationships over time. We show that our graph learning modules play the role of self-attention (Bahdanau et al., 2014), and thus our unrolled graph-based neural nets are transformers. Given the two learned graphs, we define a prediction objective for future samples in signal $\mathbf{x}$, assuming $\mathbf{x}$ is "smooth" with respect to (w.r.t.) both $\mathcal{G}^u$ and $\mathcal{G}^d$ in variational terms: *graph Laplacian regularizer* (GLR) (Pang & Cheung, 2017) for undirected $\mathcal{G}^u$, and newly designed *directed graph Laplacian regularizer* (DGLR) and *directed graph total variation* (DGTV) for directed $\mathcal{G}^d$. We devise a corresponding linear-time optimization algorithm based on *alternating direction method of multipliers* (ADMM) (Boyd et al., 2011) that unrolls into neural layers of a feed-forward network for parameter learning. **Notably, our proposed DGLR and DGTV variational terms quantify and promote signal smoothness on a directed graph, with low-pass filter interpretations**[4]. Experiments show that our

networks achieve competitive traffic forecast performance as SOTA prediction schemes, while employing drastically fewer parameters (**7.2% of transformer-based PDFormer** (Jiang et al., 2023)).

**Conflict of Interest Disclosure**   The authors declare no competing financial interests related to this work.

## 2. Preliminaries

**Undirected Graph Definitions:** Denote by $\mathcal{G}^u(\mathcal{V}, \mathcal{E}^u, \mathbf{W}^u)$ an *undirected* graph with node set $\mathcal{V} = \{1, \dots, N\}$ and undirected edge set $\mathcal{E}^u$, where $(i,j) \in \mathcal{E}^u$ implies that an undirected edge exists connecting nodes $i$ and $j$ with weight $w_{i,j}^u = W_{i,j}^u$, and $(i,j) \notin \mathcal{E}$ implies $W_{i,j}^u = 0$. $\mathbf{W}^u \in \mathbb{R}^{N \times N}$ is the symmetric *adjacency matrix* for $\mathcal{G}^u$. Denote by $\mathbf{D}^u \in \mathbb{R}^{N \times N}$ a diagonal *degree matrix*, where $D_{i,i}^u = \sum_j W_{i,j}^u$. The symmetric *graph Laplacian matrix* is $\mathbf{L}^u \triangleq \mathbf{D}^u - \mathbf{W}^u$ (Ortega et al., 2018). $\mathbf{L}^u$ is provably *positive semi-definite* (PSD) if all edge weights are non-negative $w_{i,j}^u \geq 0, \forall i, j$ (Cheung et al., 2018). The symmetric *normalized graph Laplacian* matrix is $\mathbf{L}_n^u \triangleq \mathbf{I} - (\mathbf{D}^u)^{-1/2} \mathbf{W}^u (\mathbf{D}^u)^{-1/2}$, while the asymmetric *random-walk graph Laplacian* matrix is $\mathbf{L}_r^u \triangleq \mathbf{I} - (\mathbf{D}^u)^{-1} \mathbf{W}^u$.

One can define a *graph spectrum* by eigen-decomposing the real and symmetric graph Laplacian $\mathbf{L}^u$ (or normalized graph Laplacian $\mathbf{L}_n^u$), where the $k$-th eigen-pair $(\lambda_k, \mathbf{v}_k)$ is interpreted as the $k$-th graph frequency and Fourier mode, respectively (Ortega et al., 2018). Given $\{\mathbf{v}_k\}$ are orthonormal vectors, one can decompose a signal $\mathbf{x}$ into its graph frequency components as $\boldsymbol{\alpha} = \mathbf{V}^\top \mathbf{x}$, where $\mathbf{L}^u = \mathbf{V} \text{diag}(\{\lambda_k\}) \mathbf{V}^\top$, and $\mathbf{V}^\top$ is the *graph Fourier transform* (GFT).

The $\ell_2$-norm *graph Laplacian regularizer* (GLR) (Pang & Cheung, 2017), $\mathbf{x}^\top \mathbf{L}^u \mathbf{x} = \sum_{(i,j) \in \mathcal{E}^u} w_{i,j}^u (x_i - x_j)^2$, is used to regularize an ill-posed signal restoration problem, such as denoising, to bias low-frequency signal reconstruction (*i.e.*, signals consistent with similarity graph $\mathcal{G}^u$) given observation $\mathbf{y}$:

$$\mathbf{x}^* = \arg\min_{\mathbf{x}} \|\mathbf{y} - \mathbf{x}\|_2^2 + \mu\, \mathbf{x}^\top \mathbf{L}^u \mathbf{x} = (\mathbf{I} + \mu \mathbf{L}^u)^{-1} \mathbf{y}$$

$$= \mathbf{V} \text{diag}\left(\frac{1}{1 + \mu\lambda_1}, \dots, \frac{1}{1 + \mu\lambda_N}\right) \mathbf{V}^\top \mathbf{y}. \qquad (1)$$

$\mu$ is a bias-variance tradeoff parameter, and the low-pass filter response is $f(\lambda) = (1 + \mu\lambda)^{-1}$.

**Directed Graph Definitions:**   We define analogous notations for a *directed* graph, denoted by $\mathcal{G}^d(\mathcal{V}, \mathcal{E}^d, \mathbf{W}^d)$. $[i,j] \in \mathcal{E}^d$ implies a directed edge exists from node $i$ to $j$

---

[3]A detailed review of DL models on traffic forecast is included in Appendix A.2.

[4]See a detailed review of related works on directed graph Laplacians and discussion of our novelty in directed graph frequency analysis in Appendix A.3.

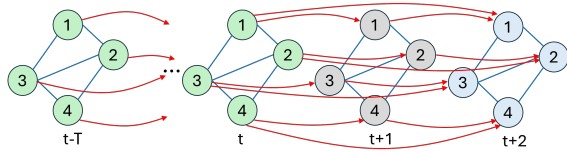

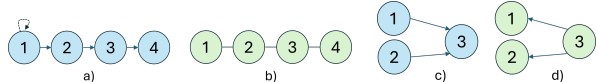

*Figure 1.* Example of a mixed graph with undirected edges (blue) connecting nodes of the same time instants, and directed edges (red) connecting nodes at $t$ to nodes in window $\{t+1, t+2\}$.

*Figure 2.* Example of a 4-node DAG $\mathcal{G}^d$, with an added self-loop at node 1, specified by $\mathbf{L}_r^d$ (a), and corresponding undirected graph $\mathcal{G}^u$, specified by $\mathcal{L}_r^d = (\mathbf{L}_r^d)^\top \mathbf{L}_r^d = \mathbf{L}^u$ (b). Example of a 3-node DAG (c), and a 3-node DAG with opposite directional edges (d).

with weight $w_{j,i}^d = W_{j,i}^d$. $\mathbf{W}^d \in \mathbb{R}^{N \times N}$ is the asymmetric *adjacency matrix*. Denote by $\mathbf{D}^d \in \mathbb{R}^{N \times N}$ a diagonal *in-degree matrix*, where $D_{i,i}^d = \sum_j W_{i,j}^d$. The *directed graph Laplacian* matrix is defined as $\mathbf{L}^d \triangleq \mathbf{D}^d - \mathbf{W}^d$. The *directed random-walk graph Laplacian* matrix is $\mathbf{L}_r^d \triangleq \mathbf{I} - (\mathbf{D}^d)^{-1}\mathbf{W}^d$.

Similar frequency notion cannot be simply defined for directed graphs via eigen-decomposition of directed graph Laplacian $\mathbf{L}^d$, because $\mathbf{L}^d$ is asymmetric, and thus the Spectral Theorem (Hawkins, 1975) does not apply. We circumvent this problem via new variational terms in Section 3.2.

## 3. Optimization Formulation & Algorithm

### 3.1. Mixed Graph for Spatial/Temporal Data

We describe the construction of a mixed graph for spatial/temporal data. A measurement station $i$ observes samples $x_i^{t-T}, \ldots, x_i^t$ at past and present instants $t - T, \ldots, t$ and samples $x_i^{t+1}, \ldots, x_i^{t+S}$ at future instants $t + 1, \ldots, t + S$. Denote by $\mathbf{x} = [\mathbf{x}^{t-T}; \ldots; \mathbf{x}^{t+S}] \in \mathbb{R}^{N(T+S+1)}$ the target graph signal—a concatenation of samples from all $N$ nodes across all $T + S + 1$ instants. A *product graph* $\mathcal{G}$ of $N \times (T + S + 1)$ nodes represents samples at $T + S + 1$ instants. $\mathcal{G}$ is a mixture of: i) undirected graph $\mathcal{G}^u$ connecting spatially node pairs of the same instants, and ii) directed graph $\mathcal{G}^d$ connecting temporally each node $i$ at instant $\tau$ to the same node at instants $\tau + 1, \ldots, \tau + W$, where $W$ denotes the pre-defined *time window*. See Fig. 1 for an illustration of a mixed graph with undirected spatial edges (blue) and directed temporal edges (red) for $W = 2$.

### 3.2. Variational Terms for Directed Graphs

Our constructed $\mathcal{G}^d$ is a *directed acyclic graph* (DAG); a directed edge always stems from a node at instant $\tau$ to a node at a future instant $\tau + s$, and thus no cycles exist. We first define a symmetric variational term for $\mathcal{G}^d$ called *directed graph Laplacian regularizer* (DGLR) to quantify variation of a signal $\mathbf{x}$ on $\mathcal{G}^d$. Denote by $\mathcal{S} \subset \mathcal{N}$ the set of *source nodes* with zero in-degrees, and $\bar{\mathcal{S}} \triangleq \mathcal{N} \setminus \mathcal{S}$ its complement. First, we add a *self-loop* of weight 1 to each node in $\mathcal{S}$; this ensures that $(\mathbf{D}^d)^{-1}$ is well-defined. Next, we define the row-stochastic *random-walk adjacency* matrix

$$\mathbf{W}_r^d \triangleq (\mathbf{D}^d)^{-1}\mathbf{W}^d.$$

#### 3.2.1. $\ell_2$-NORM VARIATIONAL TERM

Interpreting $\mathbf{W}_r^d$ as a *graph shift operator* (GSO) (Chen et al., 2015) on $\mathcal{G}^d$, we first define an $\ell_2$-norm variational term[5] for $\mathcal{G}^d$ as the squared difference between signal $\mathbf{x}$ and its graph-shifted version $\mathbf{W}_r^d\mathbf{x}$:

$$\|\mathbf{x} - \mathbf{W}_r^d\mathbf{x}\|_2^2 = \|(\mathbf{I} - \mathbf{W}_r^d)\mathbf{x}\|_2^2 = \mathbf{x}^\top \underbrace{(\mathbf{L}_r^d)^\top \mathbf{L}_r^d}_{\mathcal{L}_r^d}\mathbf{x} \quad (2)$$

where *symmetrized directed graph Laplacian* matrix $\mathcal{L}_r^d \triangleq (\mathbf{L}_r^d)^\top \mathbf{L}_r^d$ is symmetric and PSD. We call $\mathbf{x}^\top \mathcal{L}_r^d\mathbf{x}$ the DGLR, which computes the sum of squared differences between each child $j \in \bar{\mathcal{S}}$ and its parents $i$ for $[i, j] \in \mathcal{E}^d$. Note that DGLR computes to zero for the constant vector $\mathbf{1}$:

$$\mathbf{1}^\top \mathcal{L}_r^d\mathbf{1} = \mathbf{1}^\top (\mathbf{L}_r^d)^\top \mathbf{L}_r^d\mathbf{1} = \mathbf{1}^\top (\mathbf{L}_r^d)^\top (\mathbf{I} - \mathbf{W}_r^d)\mathbf{1} \overset{(a)}{=} \mathbf{0}$$

where $(a)$ follows from $\mathbf{W}_r^d$ being row-stochastic. This makes sense, as constant $\mathbf{1}$ is the smoothest signal and has no variation across any graph kernels.

We interpret eigen-pairs $\{(\xi_k, \mathbf{u}_k)\}$ of $\mathcal{L}_r^d$ as graph frequencies and graph Fourier modes for directed graph $\mathcal{G}^d$, respectively, similar to eigen-pairs $\{(\lambda_k, \mathbf{v}_k)\}$ of graph Laplacian $\mathbf{L}^u$ for an undirected graph. We prove that our frequency definition using DGLR defaults to pure sinusoids in the unweighted directed path graph case. Thus, our algorithm minimizing DGLR includes GSP schemes like Ramakrishna et al. (2020) that analyze signals along the time dimension using classical Fourier filters as special cases.

**Theorem 3.1.** *Consider a directed path graph $\mathcal{G}^d$ of $N$ nodes with directed edge weights equal to 1, where the first (source) node is augmented with a self-loop of weight 1. The symmetrized directed graph Laplacian $\mathcal{L}_r^d = (\mathbf{L}_r^d)^\top \mathbf{L}_r^d$, where $\mathbf{L}_r^d$ is the random-walk graph Laplacian for $\mathcal{G}^d$, is the same as graph Laplacian $\mathbf{L}^u$ for an undirected line graph $\mathcal{G}^u$ of $N$ nodes with edge weights equal to 1.*

See Appendix B for a formal proof.

**Remark**: Theorem 3.1 only states a special case when DGLR of a directed graph equals GLR of its corresponding

---

[5] A similar directed graph variational term was defined in Li et al. (2023) towards an objective for directed graph sampling. We focus instead on directed graph signal restoration.

undirected graph. In fact, **using our symmetrized $\mathcal{L}_r^d$ does not mean directionality has been lost.** As an example, for the 3-node DAG in Fig. 2(c), the variational term is $\|\mathbf{L}_r^d\mathbf{x}\|_2^2 = (x_3 - \frac{1}{2}(x_1 + x_2))^2$ and computes to 0 for $\mathbf{x} = [2\ 0\ 1]^\top$. On the other hand, for the DAG in Fig. 2(d) with opposite directions, the variational term is $\|\mathbf{L}_r^d\mathbf{x}\|_2^2 = (x_1 - x_3)^2 + (x_2 - x_3)^2$ and computes to 0 only if $\mathbf{x}$ is a constant vector.

### 3.2.2. $\ell_1$-NORM VARIATIONAL TERM

Using $\mathbf{W}_r^d$ again as a GSO, we define an $\ell_1$-norm variational term called *directed graph total variation* (DGTV) as

$$\|\mathbf{x} - \mathbf{W}_r^d\mathbf{x}\|_1 = \|\mathbf{L}_r^d\mathbf{x}\|_1 = \sum_{j \in \bar{\mathcal{S}}} |x_j - \sum_i w_{j,i}x_i|. \quad (3)$$

Unlike DGLR in Equation (2), DGTV is not symmetric and has no obvious frequency interpretation. Nonetheless, we provide a *two-channel filterbank* interpretation in the sequel.

### 3.3. Optimization Formulation

Denote by $\mathbf{y} \in \mathbb{R}^M$ the observation, $\mathbf{x} \in \mathbb{R}^{N(T+S+1)}$ the target signal, and $\mathbf{H} \in \{0, 1\}^{M \times N(T+S+1)}$ the sampling matrix that selects $M$ observed samples from $N(T + S + 1)$ entries in $\mathbf{x}$. Given defined undirected and directed edges, we write an objective for $\mathbf{x}$ containing a squared-error fidelity term and graph smoothness terms for the two graphs $\mathcal{G}^u$ and $\mathcal{G}^d$—GLR, DGLR, and DGTV:

$$\min_{\mathbf{x}} \|\mathbf{y} - \mathbf{H}\mathbf{x}\|_2^2 + \mu_u\mathbf{x}^\top\mathbf{L}^u\mathbf{x} + \mu_{d,2}\mathbf{x}^\top\mathcal{L}_r^d\mathbf{x} + \mu_{d,1}\|\mathbf{L}_r^d\mathbf{x}\|_1, \quad (4)$$

where $\mu_u, \mu_{d,2}, \mu_{d,1} \in \mathbb{R}_+$ are weight parameters for the three regularization terms. Combination of $\ell_2$- and $\ell_1$-norm penalties—DGLR and DGTV for directed graph $\mathcal{G}^d$ in our case[6]—is called *elastic net regularization* in statistics (Zou & Hastie, 2005) with demonstrable improved robustness. Further, employing both regularization terms means that, after algorithm unrolling, we can adapt an appropriate mixture by learning weights $\mu_{d,2}$ and $\mu_{d,1}$ per neural layer.

Minimization objective in Equation (4) is convex and composed of smooth $\ell_2$-norm terms and one non-smooth $\ell_1$-norm term. We pursue a divide-and-conquer approach and solve Equation (4) via an ADMM framework (Boyd et al., 2011) by minimizing the $\ell_2$- and $\ell_1$-norm terms alternately.

### 3.4. ADMM Optimization Algorithm

We first introduce *auxiliary variable* $\boldsymbol{\phi}$ and rewrite the optimization in Equation (4) as

$$\min_{\mathbf{x},\boldsymbol{\phi}} \|\mathbf{y} - \mathbf{H}\mathbf{x}\|_2^2 + \mu_u\mathbf{x}^\top\mathbf{L}^u\mathbf{x} + \mu_{d,2}\mathbf{x}^\top\mathcal{L}_r^d\mathbf{x} + \mu_{d,1}\|\boldsymbol{\phi}\|_1,$$
$$\text{s.t. } \boldsymbol{\phi} = \mathbf{L}_r^d\mathbf{x}. \quad (5)$$

---

[6]We can employ both $\ell_2$- and $\ell_1$-norm regularization terms for the undirected graph $\mathcal{G}^u$ as well. For simplicity, we focus on designing new regularization terms for *directed* graphs here.

Using the augmented Lagrangian method (Boyd & Vandenberghe, 2004), we rewrite Equation (5) in an unconstrained form:

$$\min_{\mathbf{x},\boldsymbol{\phi}} \|\mathbf{y} - \mathbf{H}\mathbf{x}\|_2^2 + \mu_u\mathbf{x}^\top\mathbf{L}^u\mathbf{x} + \mu_{d,2}\mathbf{x}^\top\mathcal{L}_r^d\mathbf{x} + \mu_{d,1}\|\boldsymbol{\phi}\|_1$$
$$+ \boldsymbol{\gamma}^\top(\boldsymbol{\phi} - \mathbf{L}_r^d\mathbf{x}) + \frac{\rho}{2}\|\boldsymbol{\phi} - \mathbf{L}_r^d\mathbf{x}\|_2^2 \quad (6)$$

where $\boldsymbol{\gamma} \in \mathbb{R}^{N(T+S+1)}$ is a Lagrange multiplier vector, and $\rho \in \mathbb{R}_+$ is a non-negative ADMM parameter. We solve Equation (6) iteratively by minimizing $\mathbf{x}$ and $\boldsymbol{\phi}$ in alternating steps till convergence.

### 3.4.1. MINIMIZING $\mathbf{x}^{\tau+1}$

Fixing $\boldsymbol{\phi}^\tau$ and $\boldsymbol{\gamma}^\tau$ at iteration $\tau$, optimization in Equation (6) for $\mathbf{x}^{\tau+1}$ becomes an unconstrained convex quadratic objective. The solution is a system of linear equations:

$$\left(\mathbf{H}^\top\mathbf{H} + \mu_u\mathbf{L}^u + (\mu_{d,2} + \frac{\rho}{2})\mathcal{L}_r^d\right)\mathbf{x}^{\tau+1} =$$
$$(\mathbf{L}_r^d)^\top(\frac{\rho}{2}\boldsymbol{\phi}^\tau + \frac{\boldsymbol{\gamma}^\tau}{2}) + \mathbf{H}^\top\mathbf{y}. \quad (7)$$

Because coefficient matrix $\mathbf{H}^\top\mathbf{H} + \mu_u\mathbf{L}^u + (\mu_{d,2} + \frac{\rho}{2})\mathcal{L}_r^d$ is sparse, symmetric and PD, Equation (7) can be solved in linear time via *conjugate gradient* (CG) (Shewchuk, 1994) without matrix inversion.

**Splitting $\ell_2$-norm Terms**: Instead of solving Equation (7) directly for $\mathbf{x}^{\tau+1}$ at iteration $\tau$, we introduce again auxiliary variables $\mathbf{z}_u$ and $\mathbf{z}_d$ in Equation (5) for the $\ell_2$-norm GLR and DGLR terms respectively, and rewrite the optimization as

$$\min_{\mathbf{x},\mathbf{z}_u,\mathbf{z}_d} \|\mathbf{y} - \mathbf{H}\mathbf{x}\|_2^2 + \mu_u\mathbf{z}_u^\top\mathbf{L}^u\mathbf{z}_u + \mu_{d,2}\mathbf{z}_d^\top\mathcal{L}_r^d\mathbf{z}_d$$
$$+ (\boldsymbol{\gamma}^\tau)^\top(\boldsymbol{\phi}^\tau - \mathbf{L}_r^d\mathbf{x}) + \frac{\rho}{2}\|\boldsymbol{\phi}^\tau - \mathbf{L}_r^d\mathbf{x}\|_2^2 \quad (8)$$
$$\text{s.t. } \mathbf{x} = \mathbf{z}_u = \mathbf{z}_d.$$

Using again the augmented Lagrangian method, we rewrite the unconstrained version as

$$\min_{\mathbf{x},\mathbf{z}_u,\mathbf{z}_d} \|\mathbf{y} - \mathbf{H}\mathbf{x}\|_2^2 + \mu_u\mathbf{z}_u^\top\mathbf{L}^u\mathbf{z}_u + \mu_{d,2}\mathbf{z}_d^\top\mathcal{L}_r^d\mathbf{z}_d$$
$$+ (\boldsymbol{\gamma}^\tau)^\top(\boldsymbol{\phi}^\tau - \mathbf{L}_r^d\mathbf{x}) + \frac{\rho}{2}\|\boldsymbol{\phi}^\tau - \mathbf{L}_r^d\mathbf{x}\|_2^2 + (\boldsymbol{\gamma}_u^\tau)^\top(\mathbf{x} - \mathbf{z}_u) \quad (9)$$
$$+ \frac{\rho_u}{2}\|\mathbf{x} - \mathbf{z}_u\|_2^2 + (\boldsymbol{\gamma}_d^\tau)^\top(\mathbf{x} - \mathbf{z}_d) + \frac{\rho_d}{2}\|\mathbf{x} - \mathbf{z}_d\|_2^2$$

where $\boldsymbol{\gamma}_u, \boldsymbol{\gamma}_d \in \mathbb{R}^{N(T+S+1)}$ are multipliers, and $\rho_u, \rho_d \in \mathbb{R}_+$ are ADMM parameters. Optimizing $\mathbf{x}$, $\mathbf{z}_u$ and $\mathbf{z}_d$ in Equation (9) in turn at iteration $\tau$ lead to three linear systems:

$$\left(\mathbf{H}^\top\mathbf{H} + \frac{\rho}{2}\mathcal{L}_r^d + \frac{\rho_u+\rho_d}{2}\mathbf{I}\right)\mathbf{x}^{\tau+1} = (\mathbf{L}_r^d)^\top\left(\frac{\boldsymbol{\gamma}^\tau}{2} + \frac{\rho}{2}\boldsymbol{\phi}^\tau\right)$$
$$- \frac{\boldsymbol{\gamma}_u^\tau}{2} + \frac{\rho_u}{2}\mathbf{z}_u^\tau - \frac{\boldsymbol{\gamma}_d^\tau}{2} + \frac{\rho_d}{2}\mathbf{z}_d^\tau + \mathbf{H}^\top\mathbf{y} \quad (10)$$
$$\left(\mu_u\mathbf{L}^u + \frac{\rho_u}{2}\mathbf{I}\right)\mathbf{z}_u^{\tau+1} = \frac{\boldsymbol{\gamma}_u^\tau}{2} + \frac{\rho_u}{2}\mathbf{x}^{\tau+1} \quad (11)$$
$$\left(\mu_{d,2}\mathcal{L}_r^d + \frac{\rho_d}{2}\mathbf{I}\right)\mathbf{z}_d^{\tau+1} = \frac{\boldsymbol{\gamma}_d^\tau}{2} + \frac{\rho_d}{2}\mathbf{x}^{\tau+1}. \quad (12)$$

Thus, instead of solving Equation (7) for $\mathbf{x}^{\tau+1}$ directly at iteration $\tau$, $\mathbf{x}^{\tau+1}$, $\mathbf{z}_u^{\tau+1}$ and $\mathbf{z}_d^{\tau+1}$ are optimized in turn via Equation (10) through Equation (12) using CG. This splitting induces more network parameters for data-driven learning after algorithm unrolling (see Section 4.1) towards better performance, and affords us spectral filter interpretations of the derived linear systems.

**Interpretation:** To interpret solution $\mathbf{z}_u^{\tau+1}$ spectrally, we rewrite Equation (11) as

$$\mathbf{z}_u^{\tau+1} = \left( \frac{2\mu_u}{\rho_u} \mathbf{L}^u + \mathbf{I} \right)^{-1} \left( \frac{\gamma_u}{\rho_u} + \mathbf{x}^{\tau+1} \right) \qquad (13)$$
$$= \mathbf{V} \text{diag} \left( \left\{ \frac{1}{1 + \frac{2\mu_u}{\rho_u} \lambda_k} \right\} \right) \mathbf{V}^\top \left( \frac{\gamma_u}{\rho_u} + \mathbf{x}^{\tau+1} \right)$$

where $\mathbf{L}^u = \mathbf{V} \text{diag}(\{\lambda_k\}) \mathbf{V}^\top$ is the eigen-decomposition. Thus, using frequencies defined by eigen-pairs $\{(\lambda_k, \mathbf{v}_k)\}$ of $\mathbf{L}^u$, $\mathbf{z}_u^{\tau+1}$ is the *low-pass* filter output of $\mathbf{x}^{\tau+1}$ offset by $\frac{\gamma_u}{\rho_u}$, with frequency response $f(\lambda) = (1 + \frac{2\mu_u}{\rho_u}\lambda)^{-1}$.

As done for $\mathbf{z}_u^{\tau+1}$, we can rewrite $\mathbf{z}_d^{\tau+1}$ in Equation (12) as

$$\mathbf{z}_d^{\tau+1} = \mathbf{U} \text{diag} \left( \left\{ \frac{1}{1 + \frac{2\mu_{d,2}}{\rho_d} \xi_k} \right\} \right) \mathbf{U}^\top \left( \frac{\gamma_d}{\rho_d} + \mathbf{x}^{\tau+1} \right), \quad (14)$$

where $\mathcal{L}_r^d = \mathbf{U} \text{diag}(\{\xi_k\}) \mathbf{U}^\top$ is the eigen-decomposition. Analogously, $\mathbf{z}_d^{\tau+1}$ is the *low-pass* filter output of $\mathbf{x}^{\tau+1}$ offset by $\frac{\gamma_d}{\rho_d}$, with frequency response $f(\xi) = (1 + \frac{2\mu_{d,2}}{\rho_d}\xi)^{-1}$.

Overall, alternating updates of $\mathbf{z}_u$ and $\mathbf{z}_d$ correspond to low-pass filtering under the undirected kernel $\mathbf{L}^u$ and the directed kernels $\mathcal{L}_r^d$, respectively, allowing the model to jointly capture the spatiotemporal structure encoded by the mixed graph in a principled and interpretable manner.

To interpret solution $\mathbf{x}^{\tau+1}$ in Equation (10) as a low-pass filter output, see Appendix C.

### 3.4.2. Minimizing $\boldsymbol{\phi}^{\tau+1}$

Fixing $\mathbf{x}^{\tau+1}$ at iteration $\tau$, optimizing $\boldsymbol{\phi}^{\tau+1}$ in Equation (6) means

$$\boldsymbol{\phi}^{\tau+1} = \arg\min_{\boldsymbol{\phi}} g(\boldsymbol{\phi}) \qquad (15)$$
$$= \mu_{d,1} \|\boldsymbol{\phi}\|_1 + (\boldsymbol{\gamma}^\tau)^\top (\boldsymbol{\phi} - \mathbf{L}_r^d \mathbf{x}^{\tau+1}) + \frac{\rho}{2} \|\boldsymbol{\phi} - \mathbf{L}_r^d \mathbf{x}^{\tau+1}\|_2^2.$$

We can compute Equation (15) element-wise as follows (see Appendix D for a derivation):

$$\delta = (\mathbf{L}_r^d)_i \mathbf{x}^{\tau+1} - \rho^{-1} \gamma_i^\tau,$$
$$\phi_i^{\tau+1} = \text{sign}(\delta) \cdot \max(|\delta| - \rho^{-1}\mu_{d,1}, \, 0). \qquad (16)$$

**Interpretation:** Soft-thresholding in Equation (16) to compute $\phi_i^{\tau+1}$ *attenuates* the $i$-th entry of $\mathbf{L}_r^d \mathbf{x}^{\tau+1}$. Given that graph Laplacians—second difference matrices on graphs—are high-pass filters (Ortega et al., 2018), we interpret $\mathbf{L}_r^d$

as the *high-pass channel* of a two-channel filterbank (Vetterli & Kovacevic, 2013), where the corresponding *low-pass channel* is $\mathbf{I} - \mathbf{L}_r^d = \mathbf{W}_r^d$. Using Equation (16) for processing means that the low-pass channel $\mathbf{W}_r^d$ is untouched and thus preserved. For example, the constant (low-frequency) signal $\mathbf{1}$ passes through the low-pass channel undisturbed, *i.e.*, $\mathbf{1} = \mathbf{W}_r^d \mathbf{1}$, while it is filtered out completely by the high-pass channel, *i.e.*, $\mathbf{0} = \mathbf{L}_r^d \mathbf{1}$. Thus, attenuation of the high-pass channel means Equation (16) is a low-pass filter.

### 3.4.3. Updating Lagrange Multipliers

For the objective in Equation (6), we follow standard ADMM procedure to update Lagrange multiplier $\boldsymbol{\gamma}^{\tau+1}$ after each computation of $\mathbf{x}^{\tau+1}$ and $\boldsymbol{\phi}^{\tau+1}$:

$$\boldsymbol{\gamma}^{\tau+1} = \boldsymbol{\gamma}^\tau + \rho(\boldsymbol{\phi}^{\tau+1} - \mathbf{L}_r^d \mathbf{x}^{\tau+1}). \qquad (17)$$

Splitting the $\ell_2$-norm terms means that additional multipliers $\boldsymbol{\gamma}_u^{\tau+1}$ and $\boldsymbol{\gamma}_d^{\tau+1}$ in the objective in Equation (9) must also be updated after each computation of $\mathbf{x}^{\tau+1}$, $\mathbf{z}_u^{\tau+1}$, $\mathbf{z}_d^{\tau+1}$, and $\boldsymbol{\phi}^{\tau+1}$:

$$\boldsymbol{\gamma}_u^{\tau+1} = \boldsymbol{\gamma}_u^\tau + \rho_u(\mathbf{x}^{\tau+1} - \mathbf{z}_u^{\tau+1}),$$
$$\boldsymbol{\gamma}_d^{\tau+1} = \boldsymbol{\gamma}_d^\tau + \rho_d(\mathbf{x}^{\tau+1} - \mathbf{z}_d^{\tau+1}). \qquad (18)$$

The ADMM loop is repeated till $\mathbf{x}^{\tau+1}$ and $\boldsymbol{\phi}^{\tau+1}$ converge.

# 4. Unrolled Neural Network

We unroll our iterative ADMM algorithm into an interpretable neural net. The crux to our network is two periodically inserted graph learning modules for undirected and directed edge weight computation that are akin to the classical self-attention mechanism in transformers (Bahdanau et al., 2014).

## 4.1. ADMM Algorithm Unrolling

We implement each iteration of our algorithm in Section 3.4 as a neural layer. Specifically, each ADMM iteration is implemented as four sub-layers in sequence: solving linear systems Equation (10) for $\mathbf{x}^{\tau+1}$, Equation (11) for $\mathbf{z}_u^{\tau+1}$, and Equation (12) for $\mathbf{z}_d^{\tau+1}$ via CG, and computing $\boldsymbol{\phi}^{\tau+1}$ via Equation (16). CG is itself an iterative descent algorithm, where parameters $\alpha$ and $\beta$ corresponding to step size and momentum can be tuned end-to-end (see Appendix E.1 for detailed implementation). Weight parameters $\mu_u, \mu_{d,2}, \mu_{d,1}$ for prior terms and ADMM parameters $\rho, \rho_u, \rho_d$ are also tuned per layer. Multipliers $\boldsymbol{\gamma}^{\tau+1}, \gamma_u^{\tau+1}, \gamma_d^{\tau+1}$ are then updated via Equation (17) and Equation (18) to complete one layer. Parameters learned during back-propagation for each ADMM block $b$ are denoted by $\boldsymbol{\Theta}_b$. Notably, the unrolled ADMM block acts as a low-parametric substitute for the conventional feed-forward network (FFN), and is interpretable as a set of parameterized low-pass filters over

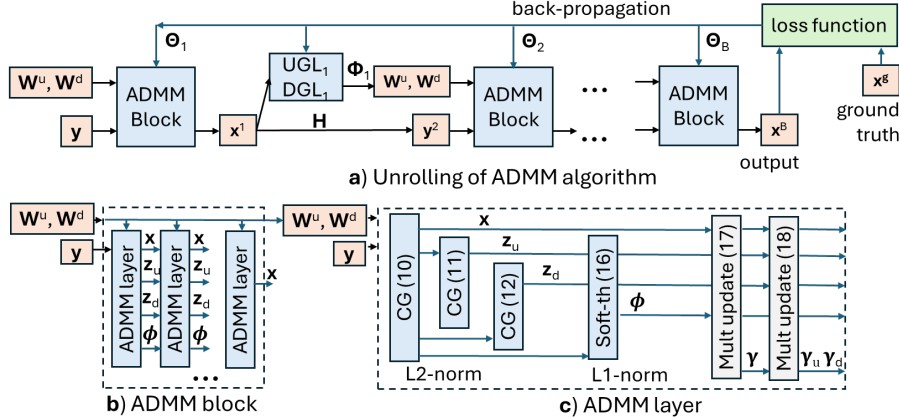

*Figure 3.* Unrolling of proposed iterative ADMM algorithm into blocks and neural layers.

the mixed graph. See Fig. 3 for an illustration, and Appendix E.2 for the detailed algorithms.

### 4.2. Self-Attention Operator in Transformer

We first review the basic self-attention mechanism: a scaled dot product following by a softmax operation (Bahdanau et al., 2014). Denote by $\mathbf{x}_i \in \mathbb{R}^E$ an *embedding* for token $i$. The *affinity* between tokens $i$ and $j$, $e(i, j)$, is the dot product of linear-transformed $\mathbf{K}\mathbf{x}_i$ and $\mathbf{Q}\mathbf{x}_j$, where $\mathbf{Q}, \mathbf{K} \in \mathbb{R}^{E \times E}$ are the *query* and *key* matrices, respectively. Using softmax, non-negative *attention weight* $a_{i,j}$ is computed from $e(i, j)$'s as

$$a_{i,j} = \frac{\exp(e(i,j))}{\sum_{l=1}^{N} \exp(e(i,l))}, \quad e(i, j) = (\mathbf{Q}\mathbf{x}_j)^{\top}(\mathbf{K}\mathbf{x}_i). \quad (19)$$

Given attention weights $a_{i,j}$, output embedding $\mathbf{y}_i$ for token $i$ is computed as

$$\mathbf{y}_i = \sum_{l=1}^{N} a_{i,l}\mathbf{x}_l \mathbf{V}, \quad (20)$$

where $\mathbf{V} \in \mathbb{R}^{E \times E}$ is a *value* matrix. *By "self-attention", we mean that the mechanism computes output embeddings by weighting a sequence's input embeddings based on internal pairwise relationships.* A transformer is thus a sequence of embedding-to-embedding mappings via self-attention operations defined by learned $\mathbf{Q}$, $\mathbf{K}$ and $\mathbf{V}$ matrices.

### 4.3. Graph Learning Modules

**Undirected Graph Learning:** We learn an undirected graph $\mathcal{G}^u$ in a module $\texttt{UGL}_b$ at block $b$. For each node $i$, we compute a low-dimensional feature vector $\mathbf{f}_i^u = F^u(\mathbf{e}_i) \in \mathbb{R}^K$, where $\mathbf{e}_i \in \mathbb{R}^E$ is a vector of signal values, position embeddings in time and Laplacian embeddings of the physical graph as done in Feng & Tassiulas (2022), and $F^u(\cdot)$ is a chosen learned nonlinear *feature function* $F^u : \mathbb{R}^E \to \mathbb{R}^K$, *e.g.*, a shallow GNN (see Appendix F.3 for details). As done

in Thuc et al. (2024), the feature distance between nodes $i$ and $j$ is computed as the *Mahalanobis distance* $\mathrm{d}^u(i, j)$:

$$\mathrm{d}^u(i, j) = (\mathbf{f}_i^u - \mathbf{f}_j^u)^{\top}\mathbf{M}(\mathbf{f}_i^u - \mathbf{f}_j^u), \quad (21)$$

where $\mathbf{M} \succeq 0$ is a symmetric PSD *metric matrix* of dimension $\mathbb{R}^{K \times K}$. Distance in Equation (21) is non-negative and *symmetric*, *i.e.*, $\mathrm{d}^u(i, j) = \mathrm{d}^u(j, i)$. Weight $w_{i,j}^u$ of undirected edge $(i, j) \in \mathcal{E}^u$ is computed as

$$w_{i,j}^u = \frac{\exp(-\mathrm{d}^u(i,j))}{\sqrt{\sum_{l \in \mathcal{N}_i} \exp(-\mathrm{d}^u(i,l))}\sqrt{\sum_{k \in \mathcal{N}_j} \exp(-\mathrm{d}^u(k,j))}}, \quad (22)$$

where the denominator summing over 1-hop neighborhood $\mathcal{N}_i$ is inserted for normalization.

**Remark**: Interpreting distance $\mathrm{d}^u(i, j)$ as $-e(i, j)$, edge weights $w_{i,j}^u$'s in Equation (22) are essentially attention weights $a_{i,j}$'s in Equation (19), and thus *the undirected graph learning module constitutes a self-attention mechanism*, albeit requiring fewer parameters, since parameters for function $F^u(\cdot)$ and metric matrix $\mathbf{M}$ can be much smaller than large and dense query and key matrices, $\mathbf{Q}$ and $\mathbf{K}$. Further, no value matrix $\mathbf{V}$ is required, since output signal $\mathbf{x}^{\tau+1}$ is computed through a set of low-pass filters—Equation (10), Equation (11), Equation (12) and Equation (16)—derived from our optimization of objective Equation (4).

**Directed Graph Learning:** Similarly, we learn a directed graph $\mathcal{G}^d$ in a module $\texttt{DGL}_b$ at block $b$. For each node $i$, we compute a feature vector $\mathbf{f}_i^d = F^d(\mathbf{e}_i) \in \mathbb{R}^K$ using feature function $F^d(\cdot)$. Like Equation (21), the Mahalanobis distance $\mathrm{d}^d(i, j)$ between nodes $i$ and $j$ is computed from $\mathbf{f}_i^d$ and $\mathbf{f}_j^d$ with a learned PSD metric matrix $\mathbf{P}$. Weight $w_{j,i}^d$ of directed edge $[i, j] \in \mathcal{E}^d$ is then computed using exponentials and normalization as

$$w_{j,i}^d = \frac{\exp(-\mathrm{d}^d(j,i))}{\sum_{(i,k) \in \mathcal{E}^d} \exp(-\mathrm{d}^d(k,i))}. \quad (23)$$

*Table 1.* Comparison of RMSE / MAE / MAPE(%) metrics of our lightweight transformer to baseline models on 30 / 60-minute forecasting in PeMS03 and METR-LA datasets. We use **boldface** for the smallest error, color the 2nd and 3nd small error in **blue**, and underline the next 2 smallest errors.

| Dataset & Horizon | PEMS03 (358 nodes, 547 edges) | | | METR-LA (207 nodes, 1,315 edges) | | |
|---|---|---|---|---|---|---|
| | 30 minutes | 60 minutes | 120 minutes | 30 minutes | 60 minutes | 120 minutes |
| VAR | 28.07 / 16.53 / 17.49 | 30.54 / 18.31 / 19.61 | 36.65 / 22.40 / 24.64 | 10.72 / 5.55 / 11.29 | 12.59 / 6.99 / 13.46 | **14.83** / 8.90 / 16.54 |
| STGCN | 28.06 / 18.24 / 17.76 | 37.31 / 24.29 / 23.00 | 54.34 / 34.83 / 30.52 | 11.26 / 5.08 / 10.93 | 13.91 / 6.35 / 13.67 | 16.92 / 8.11 / 17.69 |
| STSGCN | 26.41 / 16.75 / 16.86 | 30.62 / 19.35 / 19.15 | 38.04 / 23.86 / 23.62 | 10.25 / 4.05 / 9.18 | 12.65 / 5.18 / 11.48 | 15.74 / **6.84** / 15.33 |
| GMAN | 25.79 / 16.23 / 20.04 | 27.57 / 17.48 / 24.33 | 30.69 / 19.20 / 26.69 | 11.97 / 5.34 / 10.66 | 14.49 / 6.93 / 13.12 | 15.53 / 7.59 / 14.70 |
| ST-Wave | 25.57 / **15.11** / **15.04** | 28.65 / 16.81 / 19.24 | **29.88** / **17.11** / **17.71** | 10.81 / 4.11 / 9.16 | 13.24 / 5.33 / 11.32 | 23.18 / 11.22 / **12.71** |
| PDFormer | **23.71** / **15.05** / 18.16 | 27.16 / 17.26 / 21.21 | 35.77 / 22.25 / 25.01 | 10.21 / **3.89** / **8.50** | **12.17** / **4.81** / **10.92** | 17.27 / 9.55 / 19.10 |
| STAEformer | 30.22 / 18.85 / 26.62 | 38.36 / 23.68 / 29.21 | 48.72 / 31.96 / 44.71 | 10.16 / **3.73** / **8.25** | 12.58 / **4.79** / **10.08** | **14.63** / **5.82** / **11.87** |
| PatchSTG | **22.89** / 14.66 / 17.51 | **25.09** / **15.76** / **17.02** | **29.97** / **18.65** / 21.13 | 10.24 / **3.89** / 9.02 | 14.12 / 5.80 / 12.08 | 17.39 / 7.73 / 16.48 |
| GraphWaveNet | 25.76 / 15.22 / 16.83 | 28.15 / 18.11 / 17.56 | 34.60 / 21.10 / 22.45 | 11.42 / 5.18 / **8.21** | 12.46 / 5.17 / 11.97 | 23.13 / 11.32 / 13.53 |
| AGCRN | 27.40 / 15.19 / **14.35** | 29.90 / 16.76 / **15.32** | 32.79 / **18.72** / **17.03** | **10.11** / **3.82** / 8.61 | 12.56 / **5.00** / 11.25 | **14.77** / **6.33** / 14.05 |
| STID | 26.50 / 17.27 / 18.59 | 31.88 / 20.82 / 24.89 | 42.98 / 28.03 / 42.41 | 10.21 / 4.05 / 9.47 | 12.61 / 5.21 / 12.22 | 15.48 / 6.98 / 16.78 |
| SimpleTM | 23.75 / 15.13 / **15.59** | 25.56 / **15.97** / **15.49** | **30.58** / 18.73 / **18.18** | **8.76** / 4.12 / **7.63** | **11.96** / 5.49 / **9.65** | 15.44 / 7.64 / **12.27** |
| **Ours** | 25.05 / 15.85 / 16.49 | **26.96** / **16.58** / 18.07 | 34.06 / 20.23 / 23.86 | **10.06** / 4.05 / 9.23 | **12.17** / 5.27 / 11.78 | 15.19 / 7.14 / 16.44 |

*Table 2.* Comparison of categorized baselines and model size for 60-minute forecasting on the PEMS03 dataset against ours.

| Category | Selected Models | Params # |
|---|---|---|
| Model-based | VAR (Reinsel, 2003) | - |
| GNNs | STGCN[†](Yu et al., 2018) | 321K |
| | STSGCN (Song et al., 2020) | 3,496K |
| GATs | GMAN (Zheng et al., 2020) | 210K |
| | ST-Wave (Fang et al., 2023) | 883K |
| Transformers | PDFormer (Jiang et al., 2023) | 531K |
| | STAEformer (Liu et al., 2023) | 1,404K |
| | PatchSTG[‡](Fang et al., 2025) | 2,283K |
| Adaptive-graph models | Graph WaveNet (Wu et al., 2019) | 277K |
| | AGCRN (Bai et al., 2020) | 749K |
| MLP-based models | STID (Shao et al., 2022) | 123K |
| | SimpleTM (Chen et al., 2025) | 540K |
| **Mixed-graph unrolling** | **Ours** | **38K** |

[†] STGCN predicts one step at a time; full sequences are generated recursively.

[‡] We use the ablated PatchSTG since node coordinates are unavailable.

Parameters learned via back-propagation by the two graph learning modules at block $b$—parameters of $F^u(\cdot)$, $F^d(\cdot)$, $\mathbf{M}$ and $\mathbf{P}$—are denoted by $\mathbf{\Phi}_b$.

# 5. Experiments

## 5.1. Experimental Setup

We evaluate our model's performance on commonly used traffic speed dataset METR-LA (Li et al., 2018) and traffic flow dataset PEMS03 (Guo et al., 2022), each containing city-size traffic data with a sampling interval of 5 minutes in a local region, together with real connections and distances (or travel cost) between sensors. We reduce the original dataset size by uniformly sub-sampling 1/3 of all data[7] and split it into training, validation, and test sets by 6:2:2. We predict the traffic speed/flow in the following 30 / 60 / 120 minutes (6 / 12 / 24 steps) from the previously observed 60 minutes (12 steps) for all datasets. We restrict our model to learn a *sparse* graph based on real road connections and a limited time window to model only the *local* spatial/temporal influence. For undirected graphs $\mathcal{G}^u$, we connect each node to its $k$ nearest neighbors (kNNs). We construct 5 ADMM blocks, each containing 25 ADMM layers, and insert a pair of graph learning modules before each ADMM block. We set feature dimensions $K = 6$, and stack 4 graph learning modules vertically to learn in parallel as an implementation of the *multi-head* attention mechanism. We minimize the Huber loss with $\delta = 1$ for the *entire* reconstructed sequence and the groundtruths. See Appendix F for more details.

## 5.2. Experimental Results

We train each run of our model on an entire NVIDIA GeForce RTX 3090. We evaluate our unrolling model against the baselines shown in Table 2. Each model is trained for 70 epochs on the reduced dataset, and the metrics are the Root Mean Squared Error (RMSE), Mean Absolute Error (MAE), and Mean Absolute Percentage Error (MAPE) in the predicted 6 / 12 / 24 steps.

Table 1 shows the performance of our models and baselines in traffic forecasting, and the last column of Table 2 compares the number of parameters. We observe that our models achieve comparable performance to most baseline models, achieving top-3 performance in at least one metric for 60-minute forecasting, while employing drastically fewer parameters (**7.2% of transformer-based PDFormer**). The

[7]See Appendix G.4 for a detailed discussion on data efficiency and uniform subsampling.

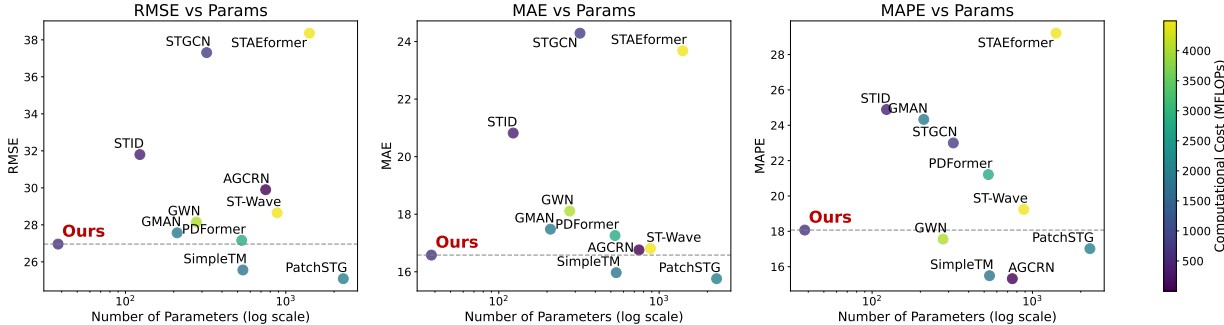

*Figure 4.* The trade-off between the number of parameters and performance for 60-minute forecast on PEMS03 dataset. The color of each scatter point represents the inference computational cost (in MFLOPs) in Table 7 in Appendix G.3.

linear model SimpleTM generally performs best on both datasets; however, it uses parameter-heavy linear models with no inductive bias, hence contains limited interpretability (Battaglia et al., 2018) and requires even more parameters than PDFormer. The largest GCN model, STSGCN, fails to achieve a satisfying performance for PEMS03, where the real connections are highly sparse. Among the largest transformers, STAEformer exhibits severe overfitting on the reduced PEMS03 dataset, whereas PatchSTG, despite achieving the best results on PEMS03, shows weaker generalization on METR-LA. We also observe that our unrolled model also generally outperforms the multi-attention model GMAN: Although both models exploit local graph neighborhoods in attention computation, our approach substantially reduces parameters by replacing conventional FFN in standard transformers with the unrolled ADMM blocks, without observable performance degradation. *In summary, while no DL-based baseline consistently dominates across both datasets and all require significantly more parameters, our model achieves top-3 performance in at least one metric for 3 of the 6 dataset–horizon settings, and remains within the top-5 in all but one.*

We plot the three metrics against parameter size (with computational cost coded by color) for 60-minute forecast on the PEMS03 datasets in Figure 4. Most baselines follow a general trend where larger models achieve lower prediction errors, except STAEFormer, which severely overfits. In contrast, our model sits distinctly in the lower-left corner in each plot, matching the accuracy of models over $10\times$ larger while using dramatically fewer parameters. Additional experiments are presented in Appendix G.

### 5.3. Interpreting the Learned Graph via Eigenvector Centrality

We analyze the relationship between the last undirected graph learned by the model and the observed traffic flows. For the best model trained on 60-minute forecast on PEMS03, we construct the learned weighted adjacency ma-

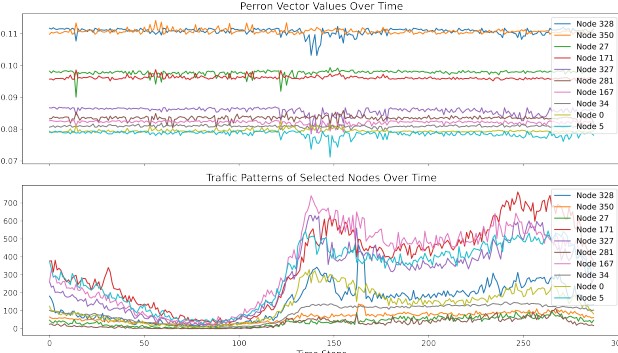

*Figure 5.* The evolution of the 10 largest Perron vector entries of the last undirected graph with the traffic flow in the first 24 hours in the test split of the PEMS03 dataset.

trices $\mathbf{W}^u$ of the undirected graphs for the first 288 time steps (24 hours) in the test set. We calculate the norm-1 eigenvector corresponding to the largest eigenvalue of $\mathbf{W}^u$, known as the *Perron vector*, which is strictly positive by the Perron-Frobenius Theorem (Horn & Johnson, 2012). The Perron vector $\mathbf{v} \in \mathbb{R}^N$ reflects the *eigenvector centrality* (Bonacich, 1987) of each node, measuring node importance through connections to other important nodes, *i.e.*, $\lambda v_i = \sum_j W_{i,j}^u v_j$ for $v_j > 0$.

Figure 5 shows the evolution of the top-10 Perron vector entries and the corresponding traffic flow. Disturbances in the Perron entries mainly occur during rapid increases in traffic flow, corresponding to the onset of morning traffic. Furthermore, the eigenvector centrality returns to the stable patterns once the traffic flow stabilizes, with little change in ranking. This indicates that **the nodes with the largest Perron vector entries remain largely consistent over time**, identifying critical congestion hubs.

To further examine short-range interactions, we next select the four most important nodes #328, #27, #350 and #171, and plot their traffic flows together with those of their physical neighbors in daytime windows of 30 and 60

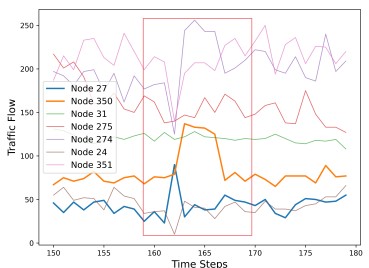
*(a)* Neighborhood of nodes #27 and #350.

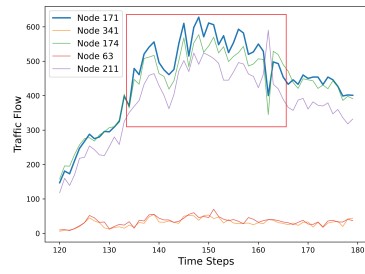
*(b)* Neighborhood of node #171.

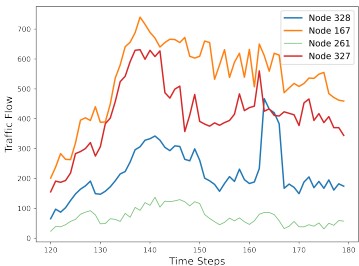
*(c)* Neighborhood of node #328.

*Figure 6.* The traffic patterns of the nodes in the neighborhood of the selected important nodes by Perron vectors.

steps in Figure 6, where the selected important nodes are highlighted with bolder lines. In the red box of Figure 6a, the spike in node #27 induces negative spikes in the neighboring nodes #274, #24 and #351; combined with the bump in node #350, this further produces the following overshoot in node #274. Figure 6b shows clear temporal delays propagating from node #171 to its neighbors, except for the "quiet" sequences. These results demonstrate that **high-centrality nodes dominate the propagation of local disturbances and temporal delays to neighboring nodes**.

Additionally, in Figure 6c, two of the three neighbors of node #328, which has the largest Perron entry, also exhibit high eigenvector centralities (#171: 5th, #167: 7th). This supports the interpretation that highly important nodes tend to connect to other important nodes. Consequently, traffic flows around node #350 are less influenced by the central node, since its neighbors themselves have relatively high centralities.

**In conclusion, our model discovers physically interpretable graph structures whose learned node importance aligns closely with real traffic dynamics.** Practically, the model can identify key intersections that critically affect traffic flow and may further guide traffic engineering in alleviating future traffic congestion.

### 5.4. Ablation Studies

We evaluate the effectiveness of the directed graph priors DGLR and DGTV, as well as the benefit of directed temporal modeling. We remove each prior individually, denoted as "w/o DGLR" and "w/o DGTV", by unrolling the corresponding ADMM algorithm. To assess directed temporal modeling, we replace the directed temporal graph $\mathcal{G}^d$ with its undirected base graph $\mathcal{G}^n$, maintaining the connectivity and weights, and substitute the directed priors with a GLR prior $\mathbf{x}^\top \mathbf{L}^n \mathbf{x}$ of the same coefficient $\mu_n = \mu_{d,2}$. Derivations of the corresponding ADMM algorithms of each ablated models are provided in Appendix I.

Table 3 shows that removing either DGLR or DGTV leads to

*Table 3.* Performance comparison in RMSE / MAE / MAPE(%) of the ablation studies for mixed-graph modeling and directed graph priors in 60-minute forecasting with PEMS03 and METR-LA.

| Model | PEMS03 | METR-LA |
|---|---|---|
| Undirected graph | 29.00 / 18.21 / 19.99 | 12.53 / 5.37 / 11.97 |
| w/o DGTV | 27.62 / 17.54 / 19.04 | 12.59 / 5.39 / 12.26 |
| w/o DGLR | 30.53 / 18.69 / 19.06 | 12.45 / 5.34 / 11.98 |
| **Ours (Mixed graph)** | **26.96 / 16.58 / 18.07** | **12.17 / 5.27 / 11.78** |

noticeable degradation in performance, which underscores their importance as directed graph priors. Moreover, our model outperforms the fully-undirected-graph-based model, indicating that sequential dependencies such as time are better modeled with directed graphs.

We further ablated the splitting-term strategy by optimizing $\mathbf{x}^{\tau+1}$ solely via Equation (7). However, the unrolled model became numerically unstable and crashed within a few epochs, suggesting that splitting the $\ell_2$ terms stabilizes training by solving Equation (10)-Equation (12) with better CG convergence. Additional ablations on the number of nearest neighbors $k$, time window size $W$, and ADMM blocks/layers are provided in Appendix H.

## 6. Conclusion

To capture complex node-to-node relations in spatial-temporal data, we unroll a mixed-graph optimization algorithm into a lightweight transformer-like neural net, where an undirected graph models spatial correlations and a directed graph models temporal relationships. We show that the two graph learning modules play the role of self-attention, meaning that our unrolled network is a transformer. We design new $\ell_2$ and $\ell_1$-norm regularizers to quantify and promote signal smoothness on directed graphs. We interpret the derived processing operations from a designed ADMM algorithm as low-pass filters. Experiments demonstrate competitive prediction performance at drastically reduced parameter counts.

## Acknowledgments

We thank the anonymous reviewers for their helpful feedback.

The work of J. Qi and H. Vicky Zhao was supported by the National Key Research and Development Program (No. 2020AAA0107800). The work of G. Cheung was supported by the Natural Sciences and Engineering Research Council of Canada (NSERC) RGPIN-2025-06252.

## Impact Statement

This paper presents work whose goal is to advance the field of Machine Learning. There are many potential societal consequences of our work, none which we feel must be specifically highlighted here.

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

# A. Related Works

## A.1. Lightweight Learning Models

Given the importance of parameter reduction for increasingly large deep learning models, there exist various studies towards this common goal across application domains. For large language models (LLMs), smaller transformer models have been achieved via low-rank assumption (Hu et al., 2022) or parameter quantization (Frantar et al., 2023). For speech enhancement, Oostermeijer et al. (2021) proposes local causal self-attention, while Zhao & Madhu (2025) proposes parameter sharing across spectral and time dimensions. For low-light image enhancement, Brateanu et al. (2025) processes the Y-channel via convolution, pooling, and multi-head self-attention, separately from the U- and V-channels. For hyperspectral image classification, Sun et al. (2024) proposes a novel multiscale 3D-2D mixed CNN design. For cervical cancer detection, Pacal (2024) substitutes larger MBConv blocks in the MaxViT architecture with smaller ConvNeXtv2 blocks, and MLP blocks with GRN-based MLPs. Similar architecture miniaturization techniques have been proposed for edge devices (Mishra & Gupta, 2024) and resource-constrained environments (Liu et al., 2024).

For traffic prediction, there are practical use cases where parameter reduction is important. For example, drivers in a city predict local traffic conditions (in an area-specific neighborhood graph) on their memory-constrained mobile devices every five minutes to compute optimal routes to their destinations, based on updated congestion information shared in their local neighborhoods, instead of sending individual requests to the cloud and overwhelming the central server.

Thuc et al. (2024) achieves a lightweight model for image interpolation in a more principled manner: first design a linear-time optimization algorithm minimizing a chosen *undirected* graph smoothness prior (such as *graph Laplacian regularizer* (GLR) (Pang & Cheung, 2017) or *graph total variation* (GTV) (Bai et al., 2019)), then unroll its iterations into neural layers to compose a "white-box" transformer-like neural net for data-driven parameter learning. This algorithm unrolling approach results in a lightweight *and* mathematically interpretable network.

## A.2. Deep Models for Traffic Forecasting

Traffic forecasting problems contain spatial and temporal correlations between stations and time steps; thus, Graph Neural Network (GNN) is naturally employed in early research. Early research, such as STGCN (Yu et al., 2018), STSGCN (Song et al., 2020) and GGRNN (Ruiz et al., 2020), employs Graph Convolutional Networks (GCNs) to capture spatial relations and uses 1D convolutions and RNNs to process temporal relationships. ASTGNN (Ta et al., 2022), Graph WaveNet (Wu et al., 2019) and AGCRN (Bai et al., 2019) explored adaptive graph settings to capture potential non-local signal relations over the graph, trading in interpretability for improved performance. Graph Attention Networks (GATs) learn graph weights from embedded signals to enable flexible and data-targeted graph learning, typical examples include GMAN (Zheng et al., 2020) and ST-Wave (Fang et al., 2023). All GNNs above focus only on the spatial correlations, while the product graph (PG) is an intuitive model that describes local spatial-temporal relations at the same time. Einizade et al. (2024) presents a continuous product graph neural network for spatial-temporal forecasting, which achieves superior performance to the previous most-popular graph methods.

Transformers extend the powerful self-attention mechanism to all spatial-temporal data points, which enhances model performance with an enormous number of parameters. Popular transformer-based models for traffic forecasting employ two transformers to process spatial and temporal dimensions separately (Xu et al., 2021; Chen et al., 2022; Huo et al., 2023), while state-of-the-art transformers introduce adaptive local attentions (Liu et al., 2023), time-delay (Jiang et al., 2023), or pre-training (Gao et al., 2024).

In recent years, simple linear models have been proposed to challenge the effectiveness and efficiency of heavy transformers. DLinear (Zeng et al., 2023) simply separates the input signals as trends and residuals, and employs 2 simple linear layers shared across nodes for prediction, but beats some of the transformer baselines in *long-term* forecasting. Other representative simple linear models include STID (Shao et al., 2022) and SimpleTM (Chen et al., 2025).

As shown in Table 2, we select two most representative and well-performing models from each category as baselines for comparison.

## A.3. Studies on Directed Graph Laplacians

In the DGSP literature, frequencies of a general graph (directed or undirected) were first defined in Sandryhaila & Moura (2014) via Jordan decomposition of the adjacency matrix, which is known to be numerically unstable. In Shafipour et al.

(2019), a set of orthornormal vectors (deemed directed graph frequencies) are computed via a minimization of spectral dispersion in a constrained quadratic programming formulation in eq. (8), which is computationally expensive. In Marques et al. (2020), a graph shift operator (GSO) $\mathbf{S}$ is assumed to be diagonalizable into $\mathbf{S} = \mathbf{V}\mathrm{diag}(\{\lambda_k\})\mathbf{V}^{-1}$, where the generally non-orthogonal eigenvectors in $\mathbf{V}$ are deemed frequency modes and ordered using a total variation measure in eq. (2), given eigenvalues $\lambda_k$'s can be complex-valued. However, asymmetric adjacency and graph Laplacian matrices are not always diagonalizable via the Spectral Theorem. Seifert & Püschel (2021) proposes to augment a directed graph with extra edges to destroy nontrivial Jordan blocks, so that the resulting adjacency matrix becomes diagonalizable. Adding edges, however, fundamentally changes the graph representation of the original pairwise relationships. Chen et al. (2023) proposes to use singular value decomposition (SVD) on the directed Laplacian to derive frequencies for directed graphs, which is computationally expensive. Kwak et al. (2024) proposes to decompose the asymmetric Laplacian matrix into a unitary part (capturing rotation or cycles) and a positive semi-definite part (capturing diffusion-like behavior) also using SVD and polar decomposition, thus suffering the same computation burden. None of these related works proposed variational terms to define directed graph frequencies, which in turn can be simply added to an objective to promote signal smoothness during optimization without first computing frequency components, as we have done.

Viswarupan et al. (2024) proposes a mixed graph representation of a digital image, where undirected edges model denoising operations, while directed edges model interpolation operations, so that together joint denoising / interpolation operators can be derived in a graph theoretical framework. In contrast, our mixed graph models specifically spatial-temporal data. In particular, we extend Thuc et al. (2024) to model the more complex node-to-node relationships in spatial-temporal data using a mixed graph: i) an *undirected* graph $\mathcal{G}^u$ to capture spatial correlations among geographically near stations, and ii) a *directed* graph $\mathcal{G}^d$ to capture sequential relationships along the temporal dimension. We define the frequency notion and promote low-pass signal reconstruction via our designed $\ell_2$-norm *directed graph Laplacian regularizer* (DGLR) variational term in Equation (2) and $\ell_1$-norm *directed graph total variation* (DGTV) variational term in Equation (3), both derived from the *graph shift operator* (GSO) (Chen et al., 2015).

Mathematically, writing signal $\mathbf{x}$ as low-pass plus high-pass, $\mathbf{x} = \mathbf{W}_r^d\mathbf{x} + \mathbf{L}_r^d\mathbf{x}$, our two regularizers suppress the same high-pass $\mathbf{L}_r^d\mathbf{x}$ but differently: the $\ell_1$ penalty $\|\mathbf{L}_r^d\mathbf{x}\|_1$ performs coordinate-wise soft-thresholding (*local* attenuation), while the $\ell_2$ penalty $\|\mathbf{L}_r^d\mathbf{x}\|_2^2$ applies spectral shrinkage according to the eigenvalues of $\mathcal{L}_r^d = (\mathbf{L}_r^d)^\top\mathbf{L}_r^d$ (*global* attenuation). Iteratively, both steer $\mathbf{x}$ toward the zero-frequency set $\{\mathbf{x}\,|\,\mathbf{x} = \mathbf{W}_r^d\mathbf{x}\}$. To the best of our knowledge, we are the first define and promote smooth signals on a directed graph in a data-driven transformer setting.

## B. Proof of Theorem 3.1

We prove Theorem 3.1 by construction as follows. A directed line graph $\mathcal{G}^d$ of $N$ nodes with inter-node edge weights $1$ (and self-loop of weight $1$ at node $1$) has adjacency matrix $\mathbf{W}_r^d$ and random-walk Laplacian matrix $\mathbf{L}_r^d = \mathbf{I}_N - \mathbf{W}_r^d$ of the following forms:

$$\mathbf{W}_r^d = \begin{bmatrix} 1 & 0 & \ldots & & 0 \\ 1 & 0 & \ldots & & 0 \\ 0 & 1 & 0 & \ldots & 0 \\ \vdots & \ddots & \ddots & & 0 \\ 0 & \ldots & 0 & 1 & 0 \end{bmatrix}, \quad \mathbf{L}_r^d = \begin{bmatrix} 0 & 0 & \ldots & & 0 \\ -1 & 1 & \ldots & & 0 \\ 0 & -1 & 1 & \ldots & 0 \\ \vdots & \ddots & \ddots & & 0 \\ 0 & \ldots & 0 & -1 & 1 \end{bmatrix}. \tag{24}$$

Note that rows 2 to $N$ of $\mathbf{L}_r^d$ actually compose the *incidence* matrix of an undirected line graph $\mathcal{G}^u$ with edge weights $1$. Thus, the symmetrized directed graph Laplacian $\mathcal{L}_r^d = (\mathbf{L}_r^d)^\top\mathbf{L}_r^d$ is also the combinatorial graph Laplacian $\mathbf{L}^u$:

$$\mathcal{L}_r^d = \mathbf{L}^u = \begin{bmatrix} 1 & -1 & 0 & \ldots & & 0 \\ -1 & 2 & -1 & 0 & \ldots & 0 \\ 0 & -1 & 2 & -1 & 0\ldots & 0 \\ \vdots & & \ddots & \ddots & \ddots & \\ 0 & \ldots & 0 & -1 & 2 & -1 \\ 0 & \ldots & & 0 & -1 & 1 \end{bmatrix}. \tag{25}$$

As an illustration, consider a 4-node directed line graph $\mathcal{G}^d$ with edge weights $1$; see Fig. 2(a) for an illustration. We see that the symmetrized directed graph Laplacian $\mathcal{L}_r^d$ defaults to the graph Laplacian $\mathbf{L}^u$ for a 4-node *undirected* line graph $\mathcal{G}^u$

(Fig. 2(b)):

$$\mathbf{W}_r^d = \begin{bmatrix} 1 & 0 & 0 & 0 \\ 1 & 0 & 0 & 0 \\ 0 & 1 & 0 & 0 \\ 0 & 0 & 1 & 0 \end{bmatrix}, \ \mathbf{L}_r^d = \begin{bmatrix} 0 & 0 & 0 & 0 \\ -1 & 1 & 0 & 0 \\ 0 & -1 & 1 & 0 \\ 0 & 0 & -1 & 1 \end{bmatrix}, \ \mathcal{L}_r^d = \begin{bmatrix} 1 & -1 & 0 & 0 \\ -1 & 2 & -1 & 0 \\ 0 & -1 & 2 & -1 \\ 0 & 0 & -1 & 1 \end{bmatrix}. \tag{26}$$

## C. Interpreting Linear System Equation (10)

For notation simplicity, we first define

$$\mathbf{r}^\tau \triangleq (\mathbf{L}_r^d)^\top \left( \frac{\boldsymbol{\gamma}^\tau}{2} + \frac{\rho}{2}\boldsymbol{\phi}^\tau \right) - \frac{\boldsymbol{\gamma}_u^\tau}{2} + \frac{\rho_u}{2}\mathbf{z}_u^\tau - \frac{\boldsymbol{\gamma}_d^\tau}{2} + \frac{\rho_d}{2}\mathbf{z}_d^\tau. \tag{27}$$

Solution $\mathbf{x}^{\tau+1}$ to linear system Equation (10) can now be written as

$$\mathbf{x}^{\tau+1} = \left( \mathbf{H}^\top\mathbf{H} + \frac{\rho}{2}\mathcal{L}_r^d + \frac{\rho_u + \rho_d}{2}\mathbf{I} \right)^{-1} (\mathbf{r}^\tau + \mathbf{H}^\top\mathbf{y}). \tag{28}$$

Define $\tilde{\mathcal{L}}_r^d \triangleq \mathbf{H}^\top\mathbf{H} + \frac{\rho}{2}\mathcal{L}_r^d$. Note that $\mathbf{H}^\top\mathbf{H}$ is a diagonal matrix with zeros and ones (corresponding to chosen sample indices) along its diagonal, and thus $\tilde{\mathcal{L}}_r^d$ is a scaled variant of $\mathcal{L}_r^d$ with the addition of selected self-loops of weight 1. Given that $\mathcal{L}_r^d = (\mathbf{L}_r^d)^\top\mathbf{L}_r^d$ is real and symmetric, $\tilde{\mathcal{L}}_r^d$ is also real and symmetric. By eigen-decomposing $\tilde{\mathcal{L}}_r^d = \tilde{\mathbf{U}}\mathrm{diag}(\{\tilde{\xi}_k\})\tilde{\mathbf{U}}^\top$, we can rewrite $\mathbf{x}^{\tau+1}$ as

$$\mathbf{x}^{\tau+1} = \tilde{\mathbf{U}}\mathrm{diag}\left( \left\{ \frac{1}{\frac{\rho_u + \rho_d}{2} + \tilde{\xi}_k} \right\} \right) \tilde{\mathbf{U}}^\top (\mathbf{r}^\tau + \mathbf{H}^\top\mathbf{y}) \tag{29}$$

which demonstrates that $\mathbf{x}^{\tau+1}$ is a low-pass filter output of up-sampled $\mathbf{H}^\top\mathbf{y}$ (with bias $\mathbf{r}^\tau$).

## D. Deriving Solution to Optimization Equation (15)

The optimization for $\boldsymbol{\phi}$ in objective Equation (15) can be performed entry-by-entry. Specifically, for the $i$-th entry $\phi_i$:

$$\min_{\phi_i} \ g(\phi_i) = \mu_{d,1}|\phi_i| + \gamma_i^\tau \left( \phi_i - (\mathbf{L}_r^d)_i\mathbf{x}^{\tau+1} \right) + \frac{\rho}{2} \left( \phi_i - (\mathbf{L}_r^d)_i\mathbf{x}^{\tau+1} \right)^2 \tag{30}$$

where $(\mathbf{L}_r^d)_i$ denotes the $i$-row of $\mathbf{L}_r^d$. $g(\phi_i)$ is convex and differentiable when $\phi_i \neq 0$. Specifically, when $\phi_i > 0$, setting the derivative of $g(\phi_i)$ w.r.t. $\phi_i$ to zero gives

$$\begin{aligned} \mu_{d,1} + \gamma_i^\tau + \rho \left( \phi_i^* - (\mathbf{L}_r^d)_i\mathbf{x}^{\tau+1} \right) &= 0 \\ \phi_i^* = (\mathbf{L}_r^d)_i\mathbf{x}^{\tau+1} - \rho^{-1}\gamma_i^\tau - \rho^{-1}\mu_{d,1} \end{aligned} \tag{31}$$

which holds only if $(\mathbf{L}_r^d)_i\mathbf{x}^{\tau+1} - \rho^{-1}\gamma_i^\tau > \rho^{-1}\mu_{d,1}$. On the other hand, when $\phi_i < 0$,

$$\begin{aligned} -\mu_{d,1} + \gamma_i^\tau + \rho \left( \phi_i^* - (\mathbf{L}_r^d)_i\mathbf{x}^{\tau+1} \right) &= 0 \\ \phi_i^* = (\mathbf{L}_r^d)_i\mathbf{x}^{\tau+1} - \rho^{-1}\gamma_i^\tau + \rho^{-1}\mu_{d,1} \end{aligned} \tag{32}$$

which holds only if $(\mathbf{L}_r^d)_i\mathbf{x}^{\tau+1} - \rho^{-1}\gamma_i^\tau < -\rho^{-1}\mu_{d,1}$. Summarizing the two cases, we get

$$\begin{aligned} \delta &= (\mathbf{L}_r^d)_i\mathbf{x}^{\tau+1} - \rho^{-1}\gamma_i^\tau \\ \phi_i^* &= \mathrm{sign}(\delta) \cdot \max(|\delta| - \rho^{-1}\mu_{d,1}, 0) \end{aligned} \tag{33}$$

**Alternative solution with subgradients:** Denote $g_1(\boldsymbol{\phi}) = \mu_{d,1}\|\boldsymbol{\phi}\|_1, g_2(\boldsymbol{\phi}) = (\boldsymbol{\gamma}^\tau)^\top(\boldsymbol{\phi} - \mathbf{L}_r^d\mathbf{x}^{\tau+1}) + \frac{\rho}{2}\|\boldsymbol{\phi} - \mathbf{L}_r^d\mathbf{x}^{\tau+1}\|_2^2$, the subgradient of $g(\boldsymbol{\phi})$ is

$$\partial g(\boldsymbol{\phi}) = \{\mathbf{g} + \nabla g_2(\boldsymbol{\phi}) | \mathbf{g} \in \partial g_1(\boldsymbol{\phi})\} = \{\mu_{d,1}\sigma + \boldsymbol{\gamma}^\tau + \rho(\boldsymbol{\phi} - \mathbf{L}_r^d\mathbf{x}^{\tau+1}) | \sigma \in \partial(\|\boldsymbol{\phi}\|_1)\}, \tag{34}$$

where

$$\partial_i(\|\boldsymbol{\phi}\|_1) = \begin{cases} \text{sign}(\phi_i), & \phi_i \neq 0, \\ [-1, 1], & \phi_i = 0. \end{cases} \tag{35}$$

The minimal $\boldsymbol{\phi}^*$ satisfies $\mathbf{0} \in \partial g(\boldsymbol{\phi}^*)$, thus

$$\begin{aligned} \mu_{d,1} + \gamma_i^\tau + \rho(\phi_i^* - (\mathbf{L}_r^d)_i \mathbf{x}) = 0, && \text{if } \phi_i^* > 0, \\ -\mu_{d,1} + \gamma_i^\tau + \rho(\phi_i^* - (\mathbf{L}_r^d)_i \mathbf{x}) = 0, && \text{if } \phi_i^* < 0, \\ \exists \sigma \in [-1, 1], \text{s.t. } \mu_{d,1}\sigma + \gamma_i^\tau + \rho(\phi_i^* - (\mathbf{L}_r^d)_i \mathbf{x}) = 0, && \text{if } \phi_i^* = 0, \end{aligned} \tag{36}$$

which gives

$$\phi_i^* = \begin{cases} (\mathbf{L}_r^d)_i \mathbf{x} - \dfrac{\gamma_i^\tau}{\rho} - \dfrac{\mu_{d,1}}{\rho}, & (\mathbf{L}_r^d)_i \mathbf{x} - \dfrac{\gamma_i^\tau}{\rho} > \dfrac{\mu_{d,1}}{\rho} \\ 0, & -\dfrac{\mu_{d,1}}{\rho} \leq (\mathbf{L}_r^d)_i \mathbf{x} - \dfrac{\gamma_i^\tau}{\rho} \leq \dfrac{\mu_{d,1}}{\rho} \\ (\mathbf{L}_r^d)_i \mathbf{x} - \dfrac{\gamma_i^\tau}{\rho} + \dfrac{\mu_{d,1}}{\rho}, & (\mathbf{L}_r^d)_i \mathbf{x} - \dfrac{\gamma_i^\tau}{\rho} < -\dfrac{\mu_{d,1}}{\rho} \end{cases} . \tag{37}$$

Thus the solution of Equation (15) is

$$\boldsymbol{\phi}^{\tau+1} = \boldsymbol{\phi}^* = \text{soft}_{\frac{\mu_{d,1}}{\rho}}\left(\mathbf{L}_r^d \mathbf{x} - \frac{\gamma_i^\tau}{\rho}\right) = \text{sign}\left(\mathbf{L}_r^d \mathbf{x} - \frac{\gamma_i^\tau}{\rho}\right) \max\left(\left|\mathbf{L}_r^d \mathbf{x} - \frac{\gamma_i^\tau}{\rho}\right| - \frac{\mu_{d,1}}{\rho}, 0\right). \tag{38}$$

## E. Algorithms for the Unrolled Network

### E.1. Parameterizing Conjugated Gradient Method in Solving Linear Equations

A linear system $\mathbf{A}\mathbf{x} = \mathbf{b}$ where $\mathbf{A}$ is a PD matrix can be solved iteratively by the conjugated gradient (CG) method, as is described in Algorithm 1. We parameterize the coefficients of the "gradients" $\alpha_k$ and "momentum" $\beta_k$ in each CG iteration to transform the iterative algorithm to stacked neural net layers. Note that $\alpha, \beta \geq 0$ in every iteration in the original algorithm, and should be kept non-negative in the unrolled version.

---

**Algorithm 1** Conjugate Gradient (CG) Method to Solve $\mathbf{A}\mathbf{x} = \mathbf{b}$ (Unrolled version $\texttt{uCG}(\mathbf{A}, \mathbf{b}, \mathbf{x}_0)$)

---

**Require:** PSD matrix $\mathbf{A} \in \mathbb{R}^{n \times n}$, vector $\mathbf{b} \in \mathbb{R}^n$, initial guess $\mathbf{x}_0$
**Ensure:** Approximate solution x such that $\|\mathbf{A}\mathbf{x} - \mathbf{b}\| \leq \epsilon$, where $\epsilon$ is the tolerance of convergence
 1: $\mathbf{r}_0 \leftarrow \mathbf{b} - \mathbf{A}\mathbf{x}_0$          ▷ *Initial residual*
 2: $\mathbf{p}_0 \leftarrow \mathbf{r}_0$
 3: $k \leftarrow 0$
 4: **while** $\|\mathbf{A}\mathbf{x}_k - \mathbf{b}\| > \epsilon$ **do**
 5:     $\boxed{\alpha_k \leftarrow \dfrac{\mathbf{r}_k^\top \mathbf{r}_k}{\mathbf{p}_k^\top \mathbf{A}\mathbf{p}_k}}$          ▷ *Learnable in unrolling*
 6:     $\mathbf{x}_{k+1} \leftarrow \mathbf{x}_k + \alpha_k \mathbf{p}_k$
 7:     $\mathbf{r}_{k+1} \leftarrow \mathbf{r}_k - \alpha_k \mathbf{A}\mathbf{p}_k$          ▷ *Dual residual*
 8:     $\boxed{\beta_k \leftarrow \dfrac{\mathbf{r}_{k+1}^\top \mathbf{r}_{k+1}}{\mathbf{r}_k^\top \mathbf{r}_k}}$          ▷ *Learnable in unrolling*
 9:     $\mathbf{p}_{k+1} \leftarrow \mathbf{r}_{k+1} + \beta_k \mathbf{p}_k$
10:     $k \leftarrow k + 1$
11: **end while**
12: **return** $\mathbf{x}_{k+1}$

---

### E.2. Detailed Algorithms and Scalability Analysis for the Proposed Model

Algorithms 2 detailedly explained the operations in ADMM blocks corresponding to Figure 3(b) and (c). An overall algorithm of the unrolled network corresponding to Figure 3(a) is shown in Algorithm 3. All matrix multiplication is

computed *locally* to ensure low computational and memory cost with

$$(\mathbf{L}^u\mathbf{x})_j = D^u_{j,j}x_j - \sum_{i \in \mathcal{N}_j} W_{i,j}x_j, \quad (\mathbf{L}^u_r\mathbf{x})_j = x_j - \sum_{i \in \mathcal{N}_i} \frac{W_{i,j}}{\sqrt{D^u_{i,i}D^u_{j,j}}}x_i, \tag{39}$$

$$(\mathbf{L}^d_r\mathbf{x})_j = x_j - \sum_{(i,j) \in \mathcal{E}^d} \frac{W^d_{j,i}}{D^d_{j,j}}x_i, \quad ((\mathbf{L}^d_r)^\top\mathbf{x})_j = x_j - \sum_{(j,i) \in \mathcal{E}^d} \frac{W^d_{i,j}}{D^d_{i,i}}x_i, \quad \mathcal{L}^d_r\mathbf{x} = (\mathbf{L}^d_r)^\top\mathbf{L}^d_r\mathbf{x}. \tag{40}$$

---

**Algorithm 2** Forward Propagation of the Unrolled `ADMM_Block`

---

**Require:** Initial signal output of the last ADMM block $\mathbf{x} \in \mathbb{R}^{T+S+1}$, undirected graph Laplacian matrix $\mathbf{L}^u$, directed graph Laplacian matrix $\mathbf{L}^d_r$.
**Ensure:** Network output of the unrolled ADMM block with $M$ layers.
    ▷ ***Learnable parameters:*** *layer-wise* $\{\mu^\tau_u, \mu^\tau_{d,1}, \mu^\tau_{d,2}, \rho^\tau, \rho^\tau_u, \rho^\tau_d\}^M_{\tau=1}$, *parameter sets* $\{(\alpha_k, \beta_k)\}$ *in each* `uCG` *module.*
1: Select real observations with mask: $\mathbf{y} \leftarrow \mathbf{Hx}, \mathbf{y} \in \mathbb{R}^{T+1}$.
2: **for** $\tau$ in $0 : M$ **do**                                                           ▷ *M ADMM layers*
3:     Initialize $\boldsymbol{\phi} \leftarrow \mathbf{L}^d_r\mathbf{x}$
4:     $\mathbf{x} \leftarrow$ `uCG`$(\mathbf{H}^\top\mathbf{H} + \frac{\rho^\tau}{2}\mathcal{L}^d_r + \frac{\rho^\tau_u+\rho^\tau_d}{2}\mathbf{I}, (\mathbf{L}^d_r)^\top(\frac{\gamma}{2} + \frac{\rho}{2}\boldsymbol{\phi}) - \frac{\gamma_u}{2} + \frac{\rho^\tau_u}{2}\mathbf{z}_u - \frac{\gamma_d}{2} + \frac{\rho^\tau_d}{2}\mathbf{z}_d + \mathbf{H}^\top\mathbf{y}, \mathbf{x})$
5:     $\mathbf{z}_u \leftarrow$ `uCG`$(\mu^\tau_u\mathbf{L}^u + \frac{\rho^\tau_u}{2}\mathbf{I}, \frac{\gamma_u}{2} + \frac{\rho^\tau_u}{2}\mathbf{x}, \mathbf{z}_u)$
6:     $\mathbf{z}_d \leftarrow$ `uCG`$(\mu^\tau_{d,2}\mathcal{L}^d_r + \frac{\rho^\tau_d}{2}\mathbf{I}, \frac{\gamma_d}{2} + \frac{\rho^\tau_d}{2}\mathbf{x}, \mathbf{z}_d)$                ▷ *Minimize* $\mathbf{x}$ *with Equations* (10) *to* (12)
7:     $\boldsymbol{\phi} \leftarrow \text{soft}_{\mu^\tau_{d,1}/\rho^\tau}(\mathbf{L}^d_r\mathbf{x} - \gamma/\rho^\tau)$                              ▷ *Minimize* $\boldsymbol{\phi}$ *with Equation* (16)
8:     $\gamma \leftarrow \gamma + \rho^\tau(\boldsymbol{\phi}^{\tau+1} - \mathbf{L}^d_r\mathbf{x}^{\tau+1})$
9:     $\gamma_u \leftarrow \gamma_u + \rho^\tau_u(\mathbf{x} - \mathbf{z}_u)$
10:    $\gamma_d \leftarrow \gamma_d + \rho^\tau_d(\mathbf{x} - \mathbf{z}_d)$                        ▷ *Update multipliers with Equations* (17) *and* (18)
11: **end for**
12: **return** $\mathbf{x}$

---

**Algorithm 3** Overall Algorithm of the Proposed Unrolled Network

---

**Require:** An observation from the training set $\mathbf{y} \in \mathbb{R}^{T+1}$.
**Ensure:** The reconstructed signal $\mathbf{x} \in \mathbb{R}^{T+S+1}$ containing both the obseration and prediction.
            ▷ ***Notations:*** *Observing stations count* $N$, *time-stamp series for the entire reconstructed signal* $\mathbf{t} \in \mathbb{R}^{T+S+1}$.
1: $\mathbf{e}_{st} \leftarrow$ `STEmb`$(N, \mathbf{t})$                      ▷ *Shared spatial-temporal embeddings (Appendix F.3.1)*
2: $\mathbf{x}^0_{\text{pred}} \leftarrow$ `Extrapolation`$(\mathbf{y}), \mathbf{x}^0_{\text{pred}} \in \mathbb{R}^S$                         ▷ *Initial prediction (Appendix F.4)*
3: $\mathbf{x} \leftarrow [\mathbf{y}; \mathbf{x}^0_{\text{pred}}], \mathbf{x} \in \mathbb{R}^{T+S+1}$
4: **for** $i$ in $0 : B$ **do**                            ▷ *B ADMM blocks and graph learning modules*
5:     $\mathbf{e}_i \leftarrow [\mathbf{x}_i; \mathbf{e}_{st,i}], \mathbf{f}_i \leftarrow F^u(\mathbf{e}_i)$ for $i = 1, \ldots, N$               ▷ *Feature extraction (Appendix F.3.2)*
6:     $\mathbf{L}^{u(h)} \leftarrow$ `UGL`$_h(\mathbf{f}_1, \ldots, \mathbf{f}_N), \quad \mathbf{L}^{d(h)}_r \leftarrow$ `DGL`$_h(\mathbf{f}_1, \ldots, \mathbf{f}_N), h = 1, 2, \ldots, H$
                                                         ▷ *Multi-head graph learning (Section 4.3)*
7:     $\mathbf{x}^{(h)}_{\text{new}} \leftarrow$ `ADMM_Block`$_h(\mathbf{x}, \mathbf{L}^{u(h)}, \mathbf{L}^{d(h)}_r), h = 1, 2, \ldots, H$
8:     $\mathbf{x}_{\text{new}} = \sum^H_{h=1} a_h\mathbf{x}^{(h)}_{\text{new}}$                              ▷ *Learnable merge of multi-head results*
9:     $\mathbf{x} \leftarrow p_i\mathbf{x}_{\text{new}} + (1 - p_i)\mathbf{x}$                      ▷ *Residual connections with learnable parameter* $p_i$
10: **end for**
11: **return** $\mathbf{x}$

---

### E.3. Scability Analysis

our model scales *linearly* as long as the city map is sparse. The conjugated gradient algorithm is detailed in Appendix E, which involves only sparse matrix-vector multiplications in each iteration. Specifically, each matrix multiplication involves graph Laplacians describing pairwise spatial and temporal similarities in local neighborhoods. Thus, as long as the temporal window $W$ and number of neighbors $k$ do not change, the computation cost in unrolled CG grows only linearly with the increase of total time steps and number of nodes. Our graph learning modules also calculate and save the edge weights

locally in each node's neighborhood, as is demonstrated in Equation (40). Taken together, with fixed $W$ and $k$, the size of our model, computation time, and memory cost all grow *linearly* with the total time steps and network size.

## F. Model Setup

### F.1. ADMM Blocks Setup

We initialize $\mu_u, \mu_{d,1}, \mu_{d,2}$ with 3, and the $\rho, \rho_u, \rho_d$ with $\sqrt{N/(T+S+1)}$, where $N$ is the number of sensors and $T+S+1$ is the full length of the recovered signal (both the observed part and the part to be predicted). Those parameters are trained every ADMM layer. We initialize the CGD parameters $(\boldsymbol{\alpha}, \boldsymbol{\beta})$s as 0.08, and tune them for every CGD iteration. We clamp each element of the CGD step size $\boldsymbol{\alpha}$ to [0, 0.8] for stability. Each element of the momentum term coefficient $\boldsymbol{\beta}$ is clamped to be non-negative.

### F.2. Graph Learning Modules Setup

We define multiple PD matrices $\mathbf{P}$s and $\mathbf{M}$s in our unrolling model due to the different impact strengths at different times. For the undirected graph, we define $\mathbf{M}^t$ as the Mahanabolis distance matrix for the undirected graph slice at time $t$ to vary the impacting strength between spatial neighbors in different time steps. For the directed graph, we assume that the impact strength varies with different *intervals* but remains uniform for different monitoring stations. Thus, we define $\mathbf{P}^w$ as the Mahanabolis distance matrix to represent the temporal influence of interval $w$ for each node. Therefore, we define $(T + S + 1)$ learnable PSD matrices $\mathbf{M}^0, \ldots, \mathbf{M}^{T+S}$ for undirected graph learning, and $W$ PSD learnable matrices $\mathbf{P}^1, \ldots, \mathbf{P}^W$ for directed graph learning in total.

We learn each PSD matrix $\mathbf{M}$ in each UGL by learning a square matrix $\mathbf{M}_0 \in \mathbb{R}^{K \times K}$ by $\mathbf{M} = \mathbf{M}_0^\top \mathbf{M}_0$, and initialize $\mathbf{M}_0$ with a matrix whose diagonal is filled with 1.5. Similarly, we learn each PSD matrix $\mathbf{P}^w$ in DGL by learning $\mathbf{P}_0^w$ where $\mathbf{P}^w = (\mathbf{P}_0^w)^\top \mathbf{P}_0^w$. We initialize $\mathbf{P}_0^w$ with $(1 + 0.2w/W)\mathbf{I}$ to set the edge weights representing influences of longer intervals slightly smaller.

We learn 4 mixed graphs in parallel with 4 sets of matrices $\mathbf{M}^0, \ldots, \mathbf{M}^{T+S}$ and $\mathbf{P}^1, \ldots, \mathbf{P}^W$ to implement the *multi-head* attention mechanism in conventional transformers. The output signals of all 4 channels are integrated to one by a linear layer at the end of each ADMM block.

### F.3. Feature Extractors Setup

We detailedly introduce our design of the feature extractor described in Section 4.3. Figure 7b shows the overall structure of our feature extractors.

#### F.3.1. INPUT EMBEDDINGS

We generate *low-parametric* spatial and temporal embeddings for each node in our mixed spatio-temporal graph, see Figure 7a. For spatial embeddings, we embed each monitoring station with a 5-dimensional learnable vector, and broadcast the embeddings to every time step in the product graph. For temporal embeddings, we follow Feng & Tassiulas (2022) to use real-time-stamp, time-in-day and day-in-week embeddings. To achieve better fitting capacity while keeping the model size small, we set only the time-in-day and day-in-week embeddings *learnable*, and use *fixed* embeddings for real time stamps.

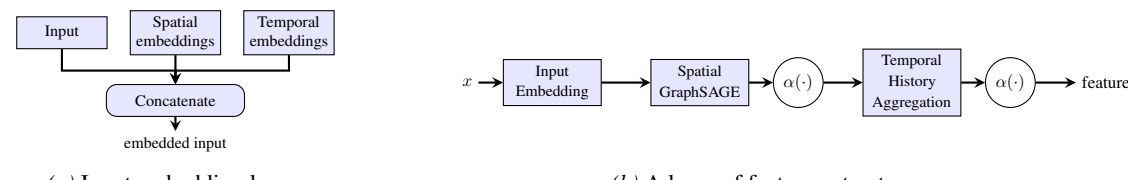

*(a)* Input embedding layer.

*(b)* A layer of feature extractor.

*Figure 7.* Framework of our feature extractor. (a) We first propose a non-parametric input embedding layer to combine signal with spatial/temporal embeddings. (b) Then aggregate embedded inputs of spatial neighbors and history information in a time window to generate a layer of feature extractor.

*Table 4.* Training setup of graph construction and batch size for all traffic datasets.

| Dataset | PeMS03 | PeMS08 | METR-LA |
|---|---|---|---|
| $k$ | 4 | 6 | 6 |
| $W$ | 6 | 6 | 6 |
| Batch size | 12 | 16 | 16 |

We use a fixed 10-dimensional sine-cosine position embeddings (Vaswani et al., 2018) for each time stamp:

$$e[t, 2i] = \sin(t/10000^i)$$
$$e[t, 2i + 1] = \cos(t/10000^i), i = 0, 1, 2, 3, 4, \tag{41}$$

and set dimensions of learnable time-in-day and day-in-week embeddings as 6 and 4. All embeddings are *shared* across all graph learning modules.

We directly concatenate these spatial and temporal embedding to our signals $\mathbf{x}_j^t$ for node $j$ at time $t$ as $\mathbf{e}_j^t = [\mathbf{x}_j^t; \mathbf{e}_j^S; \mathbf{e}_t^T]$ for feature extractors, where $\mathbf{e}_j^S$ denote the spatial Laplacian embeddings of node $j$ of the physical graph, and $\mathbf{e}_t^T$ denotes the temporal position embeddings of time step $t$.

### F.3.2. Feature Extraction from embeddings

We use a variant of GraphSAGE (Hamilton et al., 2017) to aggregate spatial features and introduce Temporal History Aggregation (THA) to aggregate spatial features. The GraphSAGE module aggregates the inputs of each node's $k$-NNs with a simple and uniformed linear layer to generate $K$-dimension spatial features for each node of each time step. The THA module aggregates the embedded inputs of past $W$ time steps for each node with a uniformed linear layer, where $W$ is the size of the time window. For both GraphSAGE and THA modules, we pad zeros to inputs as placeholders, and use Swish (Ramachandran et al., 2018) as acitvation functions with $\beta = 0.8$.

### F.4. Constructing Graphs $\mathbf{W}^{u,0}, \mathbf{W}^{d,0}$ for the First ADMM Block

Our ADMM block recovers the full signal $\mathbf{x}$ from observed signal $\mathbf{y}$ with mixed graph $\mathcal{G}^u, \mathcal{G}^d$ constructed from embedded *full* signal $\mathbf{e} = [\mathbf{x}; \mathbf{e}^S; \mathbf{e}^T]$. Thus, for the first block in ADMM, we need an *initial extrapolation* for the signals to predict.

We propose a simple module to make an initial guess of the signals: we modify the feature extractor described in Appendix F.3.2 to generate intermediate features from the embedded input of the observed part. Then we combine the feature and temporal dimensions, and use a simple linear layer with Swish activation to generate the initially extrapolated signal.

**Note:** Both the feature extractor and the initial prediction module are carefully designed to involve as few parameters as possible to make the entire model lightweight. Furthermore, low-parameter and simple feature engineering ensures that the effectiveness of our model is attributed to the unrolled blocks and graph learning modules, rather than to the trivial feature engineering parts.

### F.5. Training Setup

We preprocess the dataset by *standardizing* the entire time series for each node in space, but minimize a loss between the real groundtruth signal and the *rescaled* whole output signal. We use an Adam optimizer with learning rate $5 \times 10^{-4}$, together with a scheduler of reduction on validation loss with a reduction rate of 0.2 and patience of 5. We set different batch sizes for different datasets to leverage the most of the GPU memory (in this case, we use an entire Nvidia GeForce RTX 3090). Detailed selection of parameters such as number of nearest spatial neighbors $k$ and time window size $W$ is shown in Table 4.

## G. Further Experiments

### G.1. Additional Experimental Results

Table 5 shows the performance comparison between our model and baselines in 30-/60-minute forecast on the PEMS08 (Guo et al., 2022) dataset which contains 170 nodes and 295 edges. Results show that our model also ranks top-3 in at least

*Table 5.* Experimental results on additional dataset PEMS08 for 30- and 60-minute forecast.The indication of bolded, highlighted and underlined entries is the same as Table 1.

| Horizons | 30 minutes (6 steps) | | | 60 minutes (12 steps) | | |
|---|---|---|---|---|---|---|
| Model | RMSE | MAE | MAPE (%) | RMSE | MAE | MAPE (%) |
| VAR | 26.81 | 17.74 | 12.69 | 28.59 | 18.99 | 13.52 |
| STGCN | 36.58 | 24.39 | 16.83 | 50.61 | 33.01 | 20.57 |
| STSGCN | 26.73 | 17.43 | 11.48 | 30.22 | 19.74 | 12.90 |
| GMAN | **24.84** | **15.73** | 10.87 | **26.22** | **16.83** | 13.64 |
| ST-Wave | 26.85 | 17.19 | 11.08 | **27.28** | **16.85** | **11.77** |
| PDFormer | **22.92** | **13.91** | **9.17** | **27.36** | 17.61 | 12.90 |
| STAEFormer | 26.54 | 16.62 | 10.88 | 32.68 | 20.78 | 14.86 |
| Graph WaveNet | 30.95 | 21.65 | 13.56 | 40.24 | 29.12 | 17.79 |
| AGCRN | 34.31 | 19.91 | 10.88 | 38.09 | 22.78 | 12.75 |
| SITD | 27.40 | 18.11 | 12.10 | 33.12 | 22.66 | 20.08 |
| SimpleTM | **24.44** | **15.07** | **9.09** | 32.48 | 20.26 | **12.34** |
| **Ours** | 25.92 | 16.22 | **10.17** | 27.78 | **17.22** | **11.11** |

*Table 6.* Performance comparison of our model to the baselines on a 60-minute forecast for the entire PEMS03 dataset. The indication of bolded, highlighted and underlined entries is the same as Table 1.

| Dataset | PEMS03 | | | METR-LA | | |
|---|---|---|---|---|---|---|
| Model | RMSE | MAE | MAPE (%) | RMSE | MAE | MAPE (%) |
| VAR | 30.53 | 18.30 | 19.70 | 12.57 | 7.05 | 13.48 |
| STGCN | 36.77 | 21.81 | 19.37 | 12.27 | 5.88 | 11.79 |
| STSGCN | 28.17 | 17.46 | 16.86 | 12.73 | 5.11 | 10.66 |
| PDFormer | **25.05** | **14.78** | **15.53** | 12.32 | **4.77** | **10.20** |
| GMAN | 28.44 | 17.04 | 22.82 | 13.75 | 6.44 | 12.72 |
| GWN | 28.68 | 17.08 | 16.49 | 13.82 | 6.41 | 11.92 |
| AGCRN | 27.65 | 15.77 | **15.02** | **12.18** | **4.74** | 10.50 |
| STID | 26.80 | **15.14** | 16.45 | **12.00** | **4.63** | **10.14** |
| SimpleTM | **24.51** | **15.32** | **15.55** | 12.86 | 5.67 | **9.89** |
| **Ours** | **26.71** | 16.36 | 17.18 | **12.22** | 5.35 | 11.87 |

one metric of all experiments, and even gives the smallest MAPE among all models.

## G.2. Experiments on the Entire Datasets

To further strengthen our proposed method, we ran the 60-minute forecast experiment on the PEMS03 dataset without sampling with 70 epochs of training, and the results are shown in Table 6. Our unrolled network still ranks top-3 in at least one of the metrics for each task, indicating the consistency of our performance with subsampling.

## G.3. Inference-time Computational and Memory Cost

We compute the inference-time computational and memory cost on 60-minute forecasting for the PEMS03 dataset. We calculate the total floating-point operations (FLOPs) during inference as the computational cost with a batch size of 1, and evaluate the peak allocated memory by CUDA for each batch of size 32 as the memory cost. Results in Table 7 show that our unrolled network has the least computation cost among all the baselines except AGCRN and STID, which are generally consistently outperformed by our model. Our model's inference-time memory cost is only larger than SimpleTM and the two baselines mentioned above. In conclusion, our lightweight transformer reduces the computational cost to only 4.9% of transformer-based PDFormer and 11.2% of the GAT-based GMAN with reduced or at least comparable memory cost.

*Table 7.* Comparison of computational cost (GFLOPs) for each data point and inference memory cost (GB) for a batch of size 32 in 60-minute forecasting for the PEMS03 dataset.

| Model* | Ours | STGCN | GMAN | ST-Wave | PDFormer | STAEFormer |
|---|---|---|---|---|---|---|
| Computation (GFLOPS) | 0.087 | $0.013 \times 12^{\dagger}$ | 0.777 | 4.496 | 1.771 | 4.428 |
| Memory (GB) | 1.154 | $1.065 \times 12$ | 3.382 | 1.526 | 1.910 | 1.846 |

| Model | | PatchSTG | GWN | AGCRN | STID | SimpleTM |
|---|---|---|---|---|---|---|
| Computation (GFLOPS) | | 0.860 | $0.300 \times 12$ | 0.0002 | 0.036 | 0.670 |
| Memory (GB) | | 0.100 | $0.411 \times 12$ | 0.523 | 0.034 | 0.430 |

* The computational and memory cost for STSGCN is not included due to the limitation of applying tools such as `thop` in the public code.
$\dagger$ The $\times 12$ notation in the table means that the model predicts the outputs recursively.

*Table 8.* Performance under different sampling strides (metrics: MAE / RMSE / MAPE(%)) on 60-minute forecasting for PEMS03 dataset.

| Model | Stride = 3 (original) | Stride = 5 | Stride = 10 |
|---|---|---|---|
| **Ours** | 26.96 / 16.58 / 18.07 | 28.98 / 17.73 / 18.63 | 29.63 / 18.58 / 19.76 |
| VAR | 30.54 / 18.31 / 19.61 | 30.68 / 18.37 / 19.74 | 31.25 / 18.71 / 21.22 |
| STSGCN | 30.62 / 19.35 / 19.15 | 32.37 / 20.67 / 20.66 | 33.22 / 21.30 / 21.58 |
| PDFormer | 27.16 / 17.26 / 21.21 | 25.70 / 15.37 / 15.99 | 77.29 / 44.36 / 126.63 |
| AGCRN | 29.90 / 16.76 / 15.32 | 39.25 / 20.60 / 16.95 | 62.58 / 109.27 / 30.72 |

### G.4. Training Performance w.r.t. Training Data Cost

With a drastically smaller parameter count, our model is designed to efficiently utilize observations and generalize well under limited training data. Uniform subsampling reduces the number of training samples while preserving coverage of the entire observation sequence.

To evaluate data efficiency, we trained our models and baselines with reduced training datasets by selecting sample windows from the sequence with larger strides of 5 and 10 in the PEMS03 dataset for 60-minute forecasting. The results below show that baseline models with large parameter counts suffer from steep performance degradation when the training data becomes scarce. In contrast, our model maintains stable performance under limited data conditions, demonstrating both lower data requirements and better generalizability. This makes it particularly suitable for deployment in data-scarce or resource-constrained environments.

## H. Further Ablation Studies for Hyperparameter Sensitivity

### H.1. Sensitivity of $k, W$ in Graph Construction

We change the number of spatial nearest neighbors $k = 4, 6, 8$ and time window size $W = 4, 6, 8$ and present the results for PEMS03 and METR-LA dataset in Table 9. The selected hyperparameters are marked with "*". Decreasing or increasing $k, W$ leads to performance degradation. Our selection of $k, W$ are optimal among all test hyperparameters, considering the tradeoff with the cost of time and memory.

### H.2. Sensitivity of Number of Blocks and Layers in the Unrolling Model

We vary the number of blocks and ADMM layers for PEMS03 dataset and present the results in Table 10. The results show that reducing number of blocks or layers will increase the error in prediction. Our selected settings $n_{\text{blocks}} = 5, n_{\text{layers}} = 25$ is the optimal setting so far in consideration of the trade-off in computing memory and time cost.

*Table 9.* Performance on 60-minute forecasting with PEMS03 (left) and METR-LA (right) under different $k, W$ settings.

| | | PEMS03, 60-minute forecast | | | | | METR-LA, 60-minute forecast | | |
|---|---|---|---|---|---|---|---|---|---|
| $k$ | $W$ | RMSE | MAE | MAPE(%) | $k$ | $W$ | RMSE | MAE | MAPE(%) |
| 4* | 6* | **26.96** | **16.58** | **18.07** | 6* | 6* | **12.17** | 5.27 | **11.78** |
| 4 | 4 | 27.23 | 16.80 | 17.73 | 4 | 6 | 12.28 | 5.24 | 11.86 |
| 4 | 8 | 27.30 | 17.34 | 19.34 | 8 | 6 | 12.45 | 5.26 | 11.91 |
| 6 | 6 | 29.57 | 18.00 | 19.43 | 6 | 4 | 12.38 | **5.23** | 11.84 |
| 8 | 6 | 28.40 | 17.99 | 18.48 | 6 | 8 | 12.52 | 5.42 | 12.29 |

*Table 10.* Performance on 60-minute forecast with PEMS03 dataset under different number of blocks and layers.

| $n_{\text{blocks}}$ | $n_{\text{layers}}$ | PEMS03, 60-minute forecast | | | METR-LA, 60-minute forecast | | |
|---|---|---|---|---|---|---|---|
| | | RMSE | MAE | MAPE(%) | RMSE | MAE | MAPE(%) |
| 5* | 25* | **26.96** | **16.58** | 18.07 | **12.17** | 5.27 | **11.78** |
| 5 | 20 | 27.52 | 17.36 | 18.97 | 12.51 | 5.30 | 11.99 |
| 5 | 30 | 27.25 | 17.27 | 18.77 | 12.51 | 5.29 | 11.96 |
| 4 | 25 | 27.84 | 17.23 | **17.96** | 12.43 | 5.32 | 11.93 |
| 6 | 25 | 26.97 | 16.92 | 17.97 | 12.46 | **5.25** | 11.80 |

# I. Optimization Formulation and ADMM Algorithm in Ablation Studies

## I.1. Optimization & Algorithm without DGTV Term

By removing the DGTV term $\|\mathbf{L}_r^d \mathbf{x}\|_1$ in Equation (4), and introducing the auxiliary variables $\mathbf{z}_u, \mathbf{z}_d$ as Equation (9), we rewrite the unconstrained version of the optimization by augmented Lagrangian method as

$$\min_{\mathbf{x}, \mathbf{z}_u, \mathbf{z}_d} \|\mathbf{y} - \mathbf{H}\mathbf{x}\|_2^2 + \mu_u \mathbf{z}_u^\top \mathbf{L}^u \mathbf{z}_u + \mu_{d,2} \mathbf{z}_d^\top \mathcal{L}_r^d \mathbf{z}_d + (\boldsymbol{\gamma}_u^\tau)^\top (\mathbf{x} - \mathbf{z}_u) + \frac{\rho_u}{2}\|\mathbf{x} - \mathbf{z}_u\|_2^2$$
$$+ (\boldsymbol{\gamma}_d^\tau)^\top (\mathbf{x} - \mathbf{z}_d) + \frac{\rho_d}{2}\|\mathbf{x} - \mathbf{z}_d\|_2^2. \tag{42}$$

We optimize $\mathbf{x}, \mathbf{z}_u$ and $\mathbf{z}_d$ in turn at iteration $\tau$ as follows:

$$\left(\mathbf{H}^\top \mathbf{H} + \frac{\rho_u + \rho_d}{2}\mathbf{I}\right) \mathbf{x}^{\tau+1} = \mathbf{H}^\top \mathbf{y} - \frac{\boldsymbol{\gamma}_u^\tau + \boldsymbol{\gamma}_d^\tau}{2} + \frac{\rho_u}{2}\mathbf{z}_u^\tau + \frac{\rho_d}{2}\mathbf{z}_d^\tau, \tag{43}$$

$$\left(\mu_u \mathbf{L}^u + \frac{\rho_u}{2}\mathbf{I}\right) \mathbf{z}_u^{\tau+1} = \frac{\boldsymbol{\gamma}_u^\tau}{2} + \frac{\rho_u}{2}\mathbf{x}^{\tau+1}, \tag{44}$$

$$\left(\mu_{d,2} \mathcal{L}_r^d + \frac{\rho_d}{2}\mathbf{I}\right) \mathbf{z}_d^{\tau+1} = \frac{\boldsymbol{\gamma}_d^\tau}{2} + \frac{\rho_d}{2}\mathbf{x}^{\tau+1}. \tag{45}$$

The updating of $\boldsymbol{\gamma}_u^\tau, \boldsymbol{\gamma}_d^\tau$ follows Equation (18).

## I.2. Optimization & Algorithm without DGLR Term

By removing the DGLR term $\mathbf{x}\mathcal{L}_r^d \mathbf{x}$ in Equation (4) and introduce only the auxiliary variables $\boldsymbol{\phi}$ in Equation (5) and $\mathbf{z}_u$ in Equation (9), we rewrite the unconstrained version of the optimization of $\mathbf{x}^{\tau+1}$ by augmented Lagrangian method as

$$\min_{x, \mathbf{z}_u} \|\mathbf{y} - \mathbf{H}\mathbf{x}\|_2^2 + \mu_u \mathbf{z}_u^\top \mathbf{L}^u \mathbf{z}_u + (\boldsymbol{\gamma}^\tau)^\top (\boldsymbol{\phi}^\tau - \mathbf{L}_r^d \mathbf{x}) + \frac{\rho}{2}\|\boldsymbol{\phi}^\tau - \mathbf{L}_r^d \mathbf{x}\|_2^2 + (\boldsymbol{\gamma}_u^\tau)^\top (\mathbf{x} - \mathbf{z}_u) + \frac{\rho_u}{2}\|\mathbf{x} - \mathbf{z}_u\|_2^2 \tag{46}$$

We optimize $\mathbf{x}$ and $\mathbf{z}_u$ in turn at iteration $\tau$ as follows:

$$\left(\mathbf{H}^\top \mathbf{H} + \frac{\rho}{2}\mathcal{L}_r^d + \frac{\rho_u}{2}\mathbf{I}\right) \mathbf{x}^{\tau+1} = \mathbf{H}^\top \mathbf{y} + (\mathbf{L}_r^d)^\top (\frac{\boldsymbol{\gamma}^\tau}{2} + \frac{\rho}{2}\boldsymbol{\phi}^\tau) - \frac{\boldsymbol{\gamma}_u^\tau}{2} + \frac{\rho_u}{2}\mathbf{z}_u^\tau \tag{47}$$

$$\left(\mu_u \mathbf{L}^u + \frac{\rho_u}{2}\mathbf{I}\right) \mathbf{z}_u^{\tau+1} = \frac{\boldsymbol{\gamma}_u^\tau}{2} + \frac{\rho_u}{2}\mathbf{x}^{\tau+1} \tag{48}$$

The optimization of $\boldsymbol{\phi}^{\tau+1}$ is the same as Equation (15), and the solution is the same as Equation (16). The updating of $\boldsymbol{\gamma}, \boldsymbol{\gamma}_u$ follows Equation (17) and Equation (18).

*Table 11.* $2\sigma$ error bar of the performance metrics of our model in 60-minute traffic forecast in all three datasets.

| Dataset | PEMS03 | METR-LA | PEMS08 |
|---|---|---|---|
| $2\sigma_{\text{RMSE}}$ | 0.05 | 0.04 | 0.04 |
| $2\sigma_{\text{MAE}}$ | 0.02 | 0.02 | 0.02 |
| $2\sigma_{\text{MAPE(\%)}}$ | 0.03 | 0.04 | 0.04 |

### I.3. Algorithm Unrolling with an Undirected Temporal Graph

In this section, we are changing the directed temporal graph $\mathcal{G}^d$ into an *undirected* graph while maintaining the connections. Denote the undirected temporal graph as $\mathcal{G}^n$. We introduce the GLR operator $\mathbf{x}^\top \mathbf{L}^n \mathbf{x}$ for $\mathcal{G}^n$ on signal $\mathbf{x}$ where $\mathbf{L}^n$ is the normalized Laplacian matrix of $\mathcal{G}^n$. Thus the optimization problem is as follows:

$$\min_x \|\mathbf{y} - \mathbf{Hx}\|_2^2 + \mu_u \mathbf{x}^\top \mathbf{L}^u \mathbf{x} + \mu_n \mathbf{x}^T \mathbf{L}^n \mathbf{x} \tag{49}$$

where $\mu_u, \mu_n \in \mathbb{R}_+$ are weight parameters for the two GLR priors. The opitimization is similar to that in Appendix I.1. By a similar approach which introduces auxiliary variables $\mathbf{z}_u = \mathbf{x}, \mathbf{z}_n = \mathbf{x}$, we optimize $\mathbf{x}, \mathbf{z}_u, \mathbf{z}_n$ as follows:

$$\left(\mathbf{H}^\top \mathbf{H} + \frac{\rho_u + \rho_n}{2}\mathbf{I}\right)\mathbf{x}^{\tau+1} = \mathbf{H}^\top \mathbf{y} - \frac{\gamma_u^\tau + \gamma_n^\tau}{2} + \frac{\rho_u}{2}\mathbf{z}_u^\tau + \frac{\rho_n}{2}\mathbf{z}_n^\tau, \tag{50}$$

$$\left(\mu_u \mathbf{L}^u + \frac{\rho_u}{2}\mathbf{I}\right)\mathbf{z}_u^{\tau+1} = \frac{\gamma_u^\tau}{2} + \frac{\rho_u}{2}\mathbf{x}^{\tau+1}, \tag{51}$$

$$\left(\mu_n \mathbf{L}^n + \frac{\rho_n}{2}\mathbf{I}\right)\mathbf{z}_n^{\tau+1} = \frac{\gamma_n^\tau}{2} + \frac{\rho_n}{2}\mathbf{x}^{\tau+1}. \tag{52}$$

The updating of $\gamma_u^\tau, \gamma_n^\tau$ is also similar to Equation (18) as follows:

$$\gamma_u^{\tau+1} = \gamma_u^\tau + \rho_u(\mathbf{x}^{\tau+1} - \mathbf{z}_u^{\tau+1}), \quad \gamma_n^{\tau+1} = \gamma_n^\tau + \rho_n(\mathbf{x}^{\tau+1} - \mathbf{z}_n^{\tau+1}). \tag{53}$$

The undirected temporal graph $\mathcal{G}^n$ is learned by an `UGL` described in Section 4.3, and is initialized similar to the spatial UGL described in Appendix F.2. The initialization of $\mu_n, \rho_n$ and the CGD parameters in solving Equation (52) follows the setup in Appendix F.1.

## J. Error Bar for Our Model in Table 1

Table 11 shows the $2\sigma$ error bars of our model's performance in the 60-minute forecast in Table 1 in Section 5, each of which is computed from 3 runs on different random seeds.

## K. Limitations

One limitation of our approach is that our computed Mahalanobis distances $\mathrm{d}^u(i, j)$ and $\mathrm{d}^d(i, j)$ for respective undirected and directed graphs are non-negative (due to learned PSD metric matrices $\mathbf{M}$ and $\mathbf{P}$), whereas affinity $e(i, j)$ in traditional self-attention can be negative. We will study extensions to signed distances and more complex graph modeling as future work.

