# OpenReview forum: "Lightweight and Interpretable Transformer via Unrolling of Mixed Graph Algorithms for Traffic Forecast"
_ICML.cc/2026/Conference — ICML 2026 regular_

### Official Review · Reviewer_Q7G1 · 2026-03-02

**Soundness:** 3
**Presentation:** 2
**Significance:** 3
**Originality:** 3
**Overall Recommendation:** 4
**Confidence:** 5

**Summary:**

The paper proposes a novel transformer-like architecture for traffic forecasting that replaces the conventional black-box self-attention mechanism with an unrolled mixed-graph-based optimization algorithm. By constructing both undirected and directed graphs to represent bidirectional and downstream spatial dependencies, the authors formulate traffic forecasting as a regularized optimization problem. The resulting model, characterized by its algorithm-unrolling design, aims to provide both structural interpretability and parameter efficiency. The authors evaluate their approach on several real-world traffic benchmarks, claiming comparable performance to state-of-the-art models with significantly fewer parameters.

**Compliance With Llm Reviewing Policy:**

Affirmed.

**Final Justification:**

The comprehensive rebuttal and additional experiments effectively addressed my prior concerns regarding the method's soundness.

**Key Questions For Authors:**

- How does the convergence speed of this unrolled optimization approach compare to the training convergence of a standard learnable self-attention mechanism?

**Limitations:**

Yes

**Strengths And Weaknesses:**

**Strengths**:

- Unlike standard attention mechanisms, the unrolled layers correspond to specific optimization steps, providing a clear mathematical link between the model’s operations and graph-based spatial regularization.

- The use of mixed graphs (combining undirected and directed edges) is a sound approach to capturing the complex physical properties of traffic networks, such as flow directionality and neighborhood correlations.

**Weaknesses**:

- Although the model is highly lightweight, the quantitative results in Table 1 show that the performance improvements over existing baselines are often marginal. In several instances, the MAE and RMSE values are nearly identical to those of established models like STGNN or standard Transformers, raising concerns about whether the theoretical complexity of algorithm unrolling translates into a practical forecasting advantage beyond mere parameter reduction.

- The presentation of experimental results in Figure 4 is suboptimal. Specifically, several model names overlap with one another, making it difficult to accurately distinguish the performance of different methods. Such presentation issues hinder a precise comparative analysis and should be rectified to ensure professional clarity.

- Since the paper emphasizes "lightweight" characteristics as a primary contribution, it is vital to demonstrate that the model can maintain its efficiency and accuracy on much larger datasets (e.g., LargeST [1]). Moreover, the paper lacks a direct comparison with contemporary models specifically designed for efficiency forecasting, such as PatchSTG [2]. Comparing against PatchSTG would provide a much more meaningful benchmark for the "lightweight" claims than simply comparing against older, dense Transformers.

[1] LargeST: A Benchmark Dataset for Large-Scale Traffic Forecasting

[2] Efficient Large-Scale Traffic Forecasting with Transformers: A Spatial Data Management Perspective

---

> ### Author Rebuttal · Authors · 2026-03-31
>
> **W1.** ***The goal of our research is to “miniaturize" deep learning (DL) models while maintaining performance; thus, achieving similar performance as SOTA is a feature, not a bug.*** Beyond parameter reduction, a smaller model also means smaller storage, training and inference cost, all important metrics for real-world deployment (see Appendix A1 for a thorough discussion).
>
> Specifically for traffic forecasting, our model has demonstrated drastic parameter reduction while maintaining performance; see Figure 4 for a clear visualization. Our model has also shown substantial reduction in inference-time computation and memory cost; see Table 7 in Appendix G.3. Further, our model also demonstrated data-efficiency: when training data is scarce, our model can significantly outperform existing models (see Table 8 in Appendix G.4).
>
> Beyond empirical efficiency, our approach also offers a structural advantage that distinguishes it from all baselines. Our unrolled model achieves such dramatic parameter reduction via **optimization algorithm unrolling**: each layer in the ADMM block corresponds to a specific iteration of the ADMM algorithm, which is interpreted as low-pass graph filtering on the learned directed and undirected graph, and each graph learning model is a lightweight and graph-interpretable variant of the classical self-attention mechanism in conventional transformers. This **ante-hoc structural interpretability** means that we learn only parameters that correspond to well-defined quantities in the optimization objective, such as regularization weights and filter step sizes (the “known unknowns”), reducing parameter count. Note that this interpretability is distinct from “explainability”, that is post-hoc examination of a learned “black box” model with no attempt at parameter reduction.
>
> **W2.** We thank the reviewer for this helpful suggestion. We have revised Figure 4 to eliminate label overlaps, and the updated version is available at the following anonymous link:[ ](https://imgur.com/a/2AhQK1f)https://imgur.com/u0xNSyc. We will replace the figure accordingly in the camera-ready version.
>
> **W3.** We thank the reviewer for the comment on more extensive experiments. We have compared our model against the recommended PatchSTG below (for the setting when coordinates are unavailable). Results show that our model slightly outperforms PatchSTG on METR-LA datasets, while PatchSTG slightly outperforms ours on the PEMS03 datasets. This demonstrates that our model remains competitive against much larger recent models; our model (38K) employs only 1.7% of parameters in PatchSTG (2,283K).
>
> | Model                 | P03 30-min        | 60-min            | 120-min           | LA 30-min       | 60-min           | 120-min          |
> | --------------------- | ----------------- | ----------------- | ----------------- | --------------- | ---------------- | ---------------- |
> | Ours (38K)            | 25.05/15.85/16.49 | 26.96/16.58/18.07 | 34.06/20.23/23.86 | 10.06/4.05/9.23 | 12.17/5.27/11.78 | 15.19/7.14/16.44 |
> | PatchSTG (w/o coords) | 22.89/14.66/17.51 | 25.09/15.76/17.02 | 29.97/18.65/21.13 | 10.24/3.89/9.02 | 14.12/5.80/12.08 | 17.39/7.73/16.48 |
>
> In the limited 6-day rebuttal period, we were unable to generate comprehensive results for the much larger LargeST dataset. Given that our model scales linearly with graph size, we are confident that we can provide extensive comparison results in the camera-ready version.
>
> **A1.**  We compare validation RMSE and MAE across training epochs for our unrolled model, PDFormer (a typical transformer), and GMAN (a typical GAT) on the 60-minute forecasting task on PEMS03. We do not compare validation losses due to different training objectives (full reconstruction vs. forecasting only). As shown in the anonymous link https://imgur.com/jdeXRir, our model achieves convergence comparable to most baseline models, and the convergence is also more stable than GMAN, whose training process is terminated by early stopping. We also notice that PDFormer exhibits large validation errors in the early epochs while training errors are small, indicating that the model fits unseen data during the overfitting process on the training data.

---

> > ### Author Rebuttal · Reviewer_Q7G1 · 2026-04-04
> >
> > Thank the authors for their detailed rebuttal. Our concerns have been addressed, and we maintain our score.

---

### Official Review · Reviewer_xv1E · 2026-03-12

**Soundness:** 1
**Presentation:** 3
**Significance:** 3
**Originality:** 2
**Overall Recommendation:** 4
**Confidence:** 4

**Summary:**

This paper introduces a "white-box" transformer-like architecture for traffic forecasting derived by unrolling a mixed-graph-based optimization algorithm.

**Compliance With Llm Reviewing Policy:**

Affirmed.

**Final Justification:**

The authors have addressed my concerns

**Key Questions For Authors:**

No

**Limitations:**

Yes

**Strengths And Weaknesses:**

Strength
1) By replacing the global self-attention with localized mixed-graph operations, the model achieves a significantly lower parameter count and faster inference
2) Unlike standard black-box Transformers, the model architecture is strictly derived from a well-defined optimization problem

Weakness

1) While the paper proposes a "temporal trajectory" explanation, the added value over existing methods is questionable. Existing works already interpret predictions via receptive fields (spatial) or attention scores (importance-based).It is not clear why visualizing this specific trajectory provides more actionable insights for traffic operators.
2) The explainability is hard to be verified, since all datasets are simply representing the normal traffic pattern. Without extreme case data, it is hard to verify that the explanation here indeed can capture the traffic pattern.

---

> ### Author Rebuttal · Authors · 2026-03-31
>
> We thank the reviewer for the feedback, and we address the concerns below.
>
> **W1.** We respectfully clarify the distinction between our interpretability and existing approaches.
>
> Existing receptive-field and attention-score interpretations operate only at the **parameter level** — describing *what* the model attends to **post-hoc**, with no structural insight into *why* the model is built the way it is. In contrast, our interpretability operates at *two* connected levels, the first of which is entirely missing in existing work.
>
> In our model, at the **structural level,** each ADMM layer corresponds to a specific optimization iteration interpretable as low-pass graph filtering. This **ante-hoc** interpretability is not merely theoretical — it is precisely what enables dramatic parameter reduction (see Figure 4 for a clear illustration), since we learn only parameters corresponding to well-defined quantities in the optimization objective such as regularization weights and filter step sizes (the “known unknowns”). This paradigm of structurally interpretable networks via algorithm unrolling is well-established in the signal processing and machine learning literature (Monga et al., 2021; Thuc et al., 2024; Yu et al., 2023), and our work extends it to the spatial-temporal domain.
>
> This structural interpretability then directly enables the second level — practical interpretation of the learned graph parameters. Because the undirected graph captures spatial correlations by design, eigenvector centrality is a natural and well-justified measure of node importance, not an arbitrary visualization. The analysis in Appendix I reveals that high-centrality nodes correspond to critical congestion hubs and that the learned graph captures propagation delays between neighboring sensors (Figures 6 and 7) — directly identifying intersections that could guide traffic engineering decisions.
>
> In summary, our two-level interpretability — **structural** and **parametric** — is fundamentally distinct from existing attention-based explanations, and the structural level is entirely absent from current explainability work.
>
> **W2.** We respectfully note that the datasets are not purely static — **Figure 6** shows that our model's learned graph responds meaningfully to dynamic traffic events such as **morning rush hour spikes**, with eigenvector centrality returning to a stable state once traffic stabilizes, demonstrating interpretable behavior under changing conditions.
>
> That said, we acknowledge that extreme anomalous events such as accidents or road closures are largely absent from standard traffic benchmarks — *a limitation shared by all baseline models*. Validating interpretability under such conditions is an interesting future direction, and we will note this explicitly as a limitation in the revised paper.

---

> > ### Author Rebuttal · Reviewer_xv1E · 2026-04-04
> >
> > I have changed my recommendation to weak accept

---

> > > ### Author Response · Authors · 2026-04-05
> > >
> > > We thank the reviewer for the thoughtful re-evaluation and for acknowledging our focus on model miniaturization via algorithm unrolling. We are glad the distinction between our work and general explainability is now clear.

---

### Official Review · Reviewer_JhS9 · 2026-03-12

**Soundness:** 3
**Presentation:** 3
**Significance:** 3
**Originality:** 3
**Overall Recommendation:** 5
**Confidence:** 4

**Summary:**

The paper proposes a lightweight and interpretable transformer-like architecture for spatio-temporal traffic forecasting by unrolling an optimization algorithm into a neural network. Instead of using standard self-attention, the model learns spatial and temporal relationships through a mixed graph structure consisting of an undirected graph for spatial correlations and a directed graph for temporal dependencies. The authors formulate the prediction task as a graph-regularized optimization problem and solve it with ADMM, whose iterations are unrolled into neural network layers, providing interpretability and fewer parameters. Experiments on traffic datasets show competitive forecasting accuracy while using drastically fewer parameters than conventional transformer models.

**Compliance With Llm Reviewing Policy:**

Affirmed.

**Key Questions For Authors:**

There are severla items to be considered:
Table 1 caption contains a typo: it states “3nd”, which should be corrected to “3rd.”

The proposed model contains only 38K parameters, which is significantly lower than the parameter counts of the compared methods. It is unclear whether this parameter efficiency is an explicit objective or a key contribution of the work. If so, the authors should further justify and analyze this aspect, possibly through additional ablation studies or discussions to explain how the reduced parameter size affects performance and model design.

**Limitations:**

It is unclear how the proposed model ensures that temporal relationships are properly captured and integrated within the framework. The paper should clarify how temporality is explicitly modeled and enforced.

The explainability of the proposed framework is not sufficiently clear. Additional discussion or examples illustrating how the model provides interpretability would strengthen the paper. The baseline comparison does not fully reflect the latest state-of-the-art methods. Several compared models are relatively old (e.g., from 2018–2020). It would be beneficial to include more recent approaches to provide a stronger and more up-to-date evaluation. There are recent papers which are not compared here like this paper (Dynamic trend fusion module for traffic flow prediction, 2025 doi: https://doi.org/10.1016/j.asoc.2025.112979).  Therefore, it is recommended to bring the most recent studies.
The ablation study lacks sufficient elaboration. It is recommended that the authors explore and discuss additional aspects of the proposed framework to better demonstrate the contribution and impact of different components.

**Strengths And Weaknesses:**

Strengths
1. Clear motivation and problem relevance
The paper clearly identifies two practical limitations of conventional transformer models including large parameter size and poor interpretability, and motivates the need for lightweight and explainable architectures especially for spatio-temporal applications such as traffic forecasting.
2. Novel methodological idea
The core innovation is transforming an optimization algorithm (ADMM) for mixed spatial-temporal graphs into a neural network via algorithm unrolling, effectively replacing standard self-attention with graph-learning modules while preserving transformer-like functionality.
3. Lightweight architecture
The proposed framework significantly reduces complexity by using graph-based regularization and unrolled optimization steps, achieving competitive performance with only about 7.2% of the parameters of comparable transformer models.

Weaknesses
1. Limited performance gains
Although the model achieves competitive accuracy, the empirical results show only modest improvements over existing methods, often placing within the top few models rather than consistently outperforming them.
2. Interpretability limitations
While the architecture is more interpretable than standard transformers because each layer corresponds to an optimization step, the interpretability remains mostly theoretical and structural, and it may still be difficult to directly interpret learned graph parameters or predictions in practical scenarios.

---

> ### Author Rebuttal · Authors · 2026-03-31
>
> **W1.** We appreciate the reviewer's evaluation. To clarify: the central contribution of this paper is **model miniaturization**—achieving competitive accuracy at drastically reduced complexity—and by that metric, our results are strong (see Appendix A1 for the importance of lightweight models).
>
> Our model (38K params) is far smaller than PDFormer (531K, 7.2%) and ST-Wave (883K), while matching or outperforming them on several metrics. As is explicitly shown in Figure 4, our model sits distinctly in the lower-left corner of all 3 metric-vs-param plots, delivering the accuracy of models over 10x larger. It also reduces inference FLOPs to 4.9% of PDFormer and 11.2% of GMAN (Table 7), and under scarce training data (1/10), significantly and consistently outperforms all baselines (Table 8)—demonstrating that parameter efficiency translates directly to data efficiency.
>
> This efficiency stems from **algorithm unrolling**, where each layer corresponds to a specific ADMM iteration interpreted as low-pass graph filtering. We learn only parameters with well-defined optimization meaning (the “known unknowns”), making the reduction principled rather than arbitrary. Interpretability and efficiency are thus tightly coupled in our framework.
>
> **W2.** We appreciate this nuanced observation. We clarify that **our model's interpretability operates at two connected levels—and crucially, the first enables the second.** At the **structural level**, each ADMM layer corresponds to a specific optimization iteration interpretable as low-pass graph filtering, providing *ante-hoc* interpretability and directly enabling **practical interpretation of the learned graph parameters**. The learned undirected graph captures spatial correlations among sensors and can be analyzed via eigenvector centrality. As shown in Appendix I, high-centrality nodes correspond to key traffic hubs, and the graph further reveals meaningful patterns such as propagation delays between sensors (Figures 6 and 7). Thus, **structural and parameter-level interpretability** are mutually reinforcing and together provide insights consistent with known traffic dynamics.
>
> **A1.** We thank the reviewer for this careful comment. We will fix all the typos and grammatical issues in the camera-ready version.
>
> **A2.** Parameter efficiency is a key contribution of this work (see response to W1). Specifically, it is achieved via unrolling of the ADMM algorithm for learned graph smoothness priors, which enables us to learn only parameters that correspond to well-defined quantities in the optimization objective (see Section 5.1). For the 60-minute forecast on PEMS03, 26.0% of parameters are in the ADMM blocks, 57.5% in graph learning modules, and the remainder in feature extractors. To assess the impact of non-ADMM and non-graph-learning components, we perform ablation studies on PEMS03 (60-minute setting) as follows, showing only marginal effects on performance.
>
> |Feature dims|Feature layers|Attn heads|Metrics|
> |-|-|-|-|
> |**6**|**1**|**4**|26.96/16.58/18.07|
> |6|2|4|26.85/16.89/17.98|
> |8|1|4|28.42/17.97/19.90|
> |6|1|6|26.79/16.68/18.47|
>
> **L1.** Temporality is explicitly modeled and enforced at two levels in our framework. First, at the **graph construction level** (Section 3.1), directed edges connect each node only to its past $W$ time steps, ensuring strictly causal information flow. Second, at the **signal processing level**, minimizing the DGLR and DGTV terms (Eqs. (2)–(3)) promotes temporal smooth prediction by constraining each node’s future state to be consistent with the weighted combination of its past. In the optimization steps (Eqs. (10)-(12), (16)), the directed kernels $\mathbf{L}_r^d, \mathcal{L}_r^d$ model temporal dependencies within each node, while the undirected kernel $\mathbf{L}^u$ captures spatial correlations across nodes. Alternating low-pass filtering via these kernels enables the model to jointly capture the spatiotemporal structure in a principled and interpretable way.
>
> **L2.** We thank the reviewer and address each point in turn.
>
> *Interpretability.* Our model's interpretability operates at two levels, structural and parameter-level. See our response to W2.
>
> *Baseline recency.* We include comparisons with DSTRFormer (2025). Our model uses only 1.47% of DSTRFormer's parameters (38K vs. 2580K), while keeping the RMSE gap under 1 at the 60-minute horizon on both datasets. We note that our existing baselines were selected to represent each major architectural category (Table 2); the older models remain standard benchmarks in the traffic forecasting literature.
> |Model|P03 30-min|60-min|120-min|LA 30-min|60-min|120-min|
> |-|-|-|-|-|-|-|
> |Ours|25.05/15.85/16.49|26.96/16.58/18.07|34.06/20.23/23.86|10.06/4.05/9.23|12.17/5.27/11.78|15.19/7.14/16.44|
> |DSTRFormer|23.26/13.59/14.10|26.15/14.85/15.13|29.62/16.91/17.10|9.98/3.47/7.82|12.12/4.66/10.33|14.08/5.60/11.88|
>
> *Ablation studies.* Beyond Section 5.3, we conducted additional ablations. See our response in A2.

---

> > ### Author Rebuttal · Reviewer_JhS9 · 2026-04-03
> >
> > Thank you for the clarification. I understand your explanation; however, I believe the current scores are appropriate and accurately reflect the quality of the paper.

---

### Decision · Program_Chairs · 2026-04-30

**Decision:**

Accept (regular)

**Comment:**

This paper presents an interpretable, "white-box" alternative to the standard transformer for traffic forecasting. By unrolling an ADMM-based optimization algorithm into neural layers, the model replaces traditional self-attention with a mixed-graph framework that captures both spatial and temporal dependencies. This design provides a mathematically grounded architecture that achieves state-of-the-art accuracy with a fraction of the parameters required by conventional black-box models.

The reviewers initially raised concerns regarding the limited performance gains (JhS9, Q7G1), restricted scope of the empirical evaluation (Q7G1), and the insufficiently justified explainability (JhS9, xv1E).
In response during the rebuttal, the authors included comparisons against more recent competitors and provided an additional ablation study. Furthermore, they clarified the two-level explainability, which encompasses both structural and parameter levels. Consequently, all reviewers concurred that the rebuttal successfully addressed their initial concerns.

Therefore, I recommend acceptance. The authors are strongly encouraged to incorporate the feedback from reviewers and further strengthen the additional experimental results (e.g., adding the result on the LargeST dataset) into the final version of the manuscript.